# Coupled induction of prophage and virulence factors during tick transmission of the Lyme disease spirochete

Jenny Wachter [1,6] ✉, Britney Cheff[1], Chad Hillman[1], Valentina Carracoi[1], David W. Dorward[2], Craig Martens[3], Kent Barbian[3], Glenn Nardone[4], L. Renee Olano[4], Margie Kinnersley[5], Patrick R. Secor [5] & Patricia A. Rosa[1]

The alternative sigma factor RpoS plays a central role in the critical host-adaptive response of the Lyme disease spirochete, *Borrelia burgdorferi*. We previously identified *bbd18* as a negative regulator of RpoS but could not inactivate *bbd18* in wild-type spirochetes. In the current study we employed an inducible *bbd18* gene to demonstrate the essential nature of BBD18 for viability of wild-type spirochetes in vitro and at a unique point in vivo. Transcriptomic analyses of BBD18-depleted cells demonstrated global induction of RpoS-dependent genes prior to lysis, with the absolute requirement for BBD18, both in vitro and in vivo, circumvented by deletion of *rpoS*. The increased expression of plasmid prophage genes and the presence of phage particles in the supernatants of lysing cultures indicate that RpoS regulates phage lysis-lysogeny decisions. Through this work we identify a mechanistic link between endogenous prophages and the RpoS-dependent adaptive response of the Lyme disease spirochete.

*Borrelia burgdorferi*, the etiological agent of Lyme disease, is an obligate symbiont that must transition between an *Ixodes* tick vector and competent vertebrate hosts during its enzootic cycle[1-3]. To colonize these disparate environments, *B. burgdorferi* tightly regulates gene expression and protein profile in response to environmental cues. Intake of a blood meal by feeding ticks leads to dramatic changes in ambient temperature, nutrient availability, osmolarity, and pH of the tick midgut[4-6]. Spirochetes colonizing the tick midgut sense and respond to the environmental variables that accompany tick feeding through induction of a regulatory cascade that results in characteristic changes in gene expression[7-12] and rapid expansion of the spirochete population in the tick midgut[13-15]. This adaptive response, which requires the alternative sigma factor RpoS, is a prerequisite for host infection following transmission of the Lyme disease spirochete by the feeding tick vector[16-19].

Efforts to characterize the RpoS-dependent regulatory cascade of *B. burgdorferi* have led to the identification of approximately 100 differentially expressed genes[16,17,20-25]. While RpoS is critical for mammalian infection, it is not required by spirochetes in vitro[16,26-28], and is only expressed when culture conditions are manipulated to simulate changes associated with tick feeding[6,26,29-31]. Conversely, over-expression of *rpoS* is lethal for in vitro-grown spirochetes[32,33]. Chen and colleagues observed that higher expression of *rpoS* in *B. burgdorferi* led to profound membrane blebbing and disruption, culminating in cell lysis[33]. While it is unclear why over-expression of RpoS is detrimental to the spirochete, it is possible that unnaturally high levels of RpoS outcompete other sigma factors for a limited pool of core RNA polymerase, resulting in inadequate expression of essential housekeeping genes[34].

[1]Laboratory of Bacteriology, Rocky Mountain Laboratories, National Institute of Allergy and Infectious Diseases, National Institutes of Health, Hamilton, MT, USA. [2]Electron Microscopy Unit, Research Technologies Branch, Rocky Mountain Laboratories, National Institutes of Allergy and Infectious Diseases, National Institutes of Health, Hamilton, MT, USA. [3]Genomics Unit, Research Technologies Branch, Rocky Mountain Laboratories, National Institute of Allergy and Infectious Diseases, National Institutes of Health, Hamilton, MT, USA. [4]Protein Chemistry Section, Research Technologies Branch, National Institute of Allergy and Infectious Diseases, National Institutes of Health, Rockville, MD, USA. [5]Division of Biological Sciences, The University of Montana, Missoula, MT, USA. [6]Present address: Vaccine and Infectious Disease Organization, University of Saskatchewan, Saskatoon, SK, Canada. ✉e-mail: jenny.wachter@usask.ca

The *Borrelia* genome is highly segmented, composed of a linear chromosome and more than 20 distinct and co-existing linear and circular plasmids[35–40]. In the widely used B31 type strain, three distinct linear plasmids and a family of ~10 closely related circular plasmids, termed cp32s, encode prophages with genes that are positively regulated by RpoS[24,25,41–44]. The endogenous cp32 prophage are conserved amongst all Lyme disease *Borrelia* and have been shown to mediate horizontal gene transfer in vitro through transduction of plasmid and chromosomal DNA[45,46]. Additionally, prophage virions have been detected in association with environmental and clinical isolates of *B. burgdorferi*, but their genetic identities have not been defined[47,48]. This knowledge, combined with the fact that the Lyme disease spirochete exists in nature as a heterologous population of closely related bacteria with sufficient antigenic diversity for co-infection and re-infection of immune hosts[49–54], and the well-documented evidence of genetic diversity for (plasmid-encoded) genes through horizontal gene transfer, lends credence to the occurrence of transduction in nature. However, the ecological niche and mechanism of DNA exchange between Lyme disease spirochetes have not been defined[55–59].

In this work we report a link between endogenous prophages and the BBD18-modulated, RpoS-dependent adaptive response of the Lyme disease spirochete during tick feeding. We previously identified the plasmid-encoded *bbd18* gene as a negative regulator of the global RpoS-dependent adaptive response of *B. burgdorferi* but failed to inactivate *bbd18* in a wild-type B31 background[60–62]. In the current study we utilize an inducible *bbd18* construct, genome-wide transcriptomic and protein profiling, and engineered displacement of all cp32 prophage plasmids, to investigate the lethal phenotype that accompanies BBD18 depletion and unregulated production of RpoS in infectious *B. burgdorferi*. We analyze the global transcriptomes of spirochetes lacking BBD18 and RpoS to gain insight into the mechanisms governing induction of the RpoS regulon. Additionally, we query whether lethality accompanies BBD18 depletion in vivo and identify the precise stage and location during the mouse-tick infectious cycle at which spirochetes require BBD18-mediated modulation of RpoS for survival. We propose that the segmented nature of the *Borrelia* genome facilitates RpoS-dependent, phage-mediated horizontal transfer of key plasmid-borne genes in the tick midgut prior to transmission.

## Results

### Phenotype of inducible *bbd18* cells in culture

Previous attempts to inactivate *bbd18* in wild-type (wt) infectious *B. burgdorferi* have been unsuccessful[61]. Therefore, we chose to employ a conditional mutant to investigate the phenotype of spirochetes lacking BBD18. To this end, we created a strain (termed flacp::ibbd18) through allelic exchange with an IPTG-inducible copy of the *flacp*-driven *bbd18* gene at the native locus on linear plasmid 17 (lp17) in infectious clone B31-68-LS (wt), which encodes *lacI* on lp25 (Fig. 1a, b)[61,63,64]. PCR and sequencing confirmed the structure of the IPTG-inducible *bbd18* locus (*ibbd18*) on lp17 in flacp::ibbd18 spirochetes. As expected, addition or subtraction of IPTG to the wt strain did not affect *bbd18* transcripts, while addition of increasing amounts of IPTG to flacp::ibbd18 cultures resulted in concomitant increase in *bbd18* transcripts (Fig. 1c). Notably, removal of IPTG from flacp::ibbd18 cultures resulted in minimal transcription of *bbd18*. However, at all IPTG concentrations analyzed, *bbd18* transcript levels in flacp::ibbd18 cells were lower than wt. Despite the lower level of inducible *ibbd18* expression, growth of flacp::ibbd18 was indistinguishable from wt at IPTG concentrations of 0.1 mM or higher (Fig. 1d). Consistent with previous failed attempts to delete *bbd18* in wt *B. burgdorferi*, spirochetes carrying the inducible *ibbd18* gene

exhibited an IPTG-dependent growth phenotype, with <10% of the starting population remaining by 48 hours after removal of IPTG (Fig. 1d), and few intact spirochetes detected by live/dead staining (Fig. 2), indicating that de-induction of *ibbd18* resulted in cell lysis. However, approximately 4 days following removal of IPTG from flacp::ibbd18 cultures, motile cells could again be visualized by dark-field microscopy, followed by growth to wt level (Fig. 1d). The IPTG-independent growth phenotype of this recovered population, designated flacp::ibbd18(rec), was genetically stable and indistinguishable in growth rate from wt when passed to fresh medium without IPTG, consistent with the existence of a suppressor mutation that eliminates conditional lethality in flacp::ibbd18(rec).

### Genetic basis for growth phenotype of flacp:ibbd18(rec) cells

We reasoned that these flacp::ibbd18(rec) cells might provide some insight into the regulation or mechanism of cell death that accompanies BBD18 depletion in wt spirochetes. To identify the putative suppressor mutation(s) that allow flacp::ibbd18(rec) to grow without IPTG induction, long-read PacBio sequencing was performed on total genomic DNA isolated from isogenic wt, flacp::ibbd18, a non-clonal flacp::ibbd18(rec) outgrowth, and its clonal derivatives (Supplementary Data 1: PacBio). This analysis revealed mutations in the *lacI* gene on lp25 of ibbd18(rec), but not elsewhere in the genome, suggesting an inability of the mutated LacI (LacI*) repressor to bind to the *lacO* operator sequence in the *flacp* promoter of flacp::ibbd18(rec). Quantitative reverse transcriptase PCR (qRT-PCR) (Fig. 1c) indicated that transcription of *ibbd18* from *flacp* was no longer repressed by the mutated LacI* in flacp::ibbd18(rec) cells. This is not our first encounter with escape mutants from the *flacp*-inducible system in *B. burgdorferi*; we previously reported mutations in the *lacO* operator sequence of *flacp* that prevented LacI-mediated repression of the essential *ftsH* gene of *B. burgdorferi*[63]. The observed selection for suppressor mutations that allow unconditional expression of *ibbd18* in flacp::ibbd18(rec) cells further illustrates the requirement for BBD18 by *B. burgdorferi* in culture, but does not elucidate its function or the lethal sequela that accompany depletion of BBD18 in wt spirochetes.

### RNA-sequence and mass-spectrometry analyses of flacp::ibbd18 cells

We next performed global proteomic and transcriptomic analyses to gain insight into the phenotype of de-induced flap::ibbd18 cells prior to lysis. RNA-sequencing (RNA-seq) and tandem mass tag mass-spectrometry (TMT-MS) analyses were performed on wt and flacp::ibbd18 cells and flacp::ibbd18(rec) grown in the presence (induced) and absence (de-induced) of IPTG. Similar to qRT-PCR data, *bbd18*/BBD18 was expressed at induced levels in flacp::ibbd18(rec) cells cultured in the absence of IPTG, confirming the expression of *bbd18*/BBD18 in these recovery cells as a result of a suppressor mutation (Supplementary Data 1: up-regulated, down-regulated, and TMT-MS). Significantly, the transcriptome of induced flacp::ibbd18 cells closely resembled that of wt cells, with only 25 genes exhibiting increased expression, while one gene, *bbd18*, and one small RNA, SR0160, had decreased expression (Supplementary Data 1: up-regulated and down-regulated & Supplemental Fig. S1). 21 of the 25 up-regulated genes belong to the previously defined RpoS-regulon[24,25]. The 4 up-regulated genes that are not part of the identified RpoS-regulon include *bbj25* on lp38, which encodes a hypothetical protein; *bbe31* on lp25, which encodes an outer surface protein; and *bbd20* and *bbd21* on lp17, which encode a pseudogene and a hypothetical protein, respectively. We conclude that the sub-wildtype level of *bbd18* expression in induced flacp::ibbd18 cells resulted in a partially RpoS-induced state and upregulation of a limited set of genes.

In contrast with the above, and consistent with the lethal phenotype that follows unmodulated RpoS induction, the global transcriptome of de-induced flacp::ibbd18 varied dramatically relative to

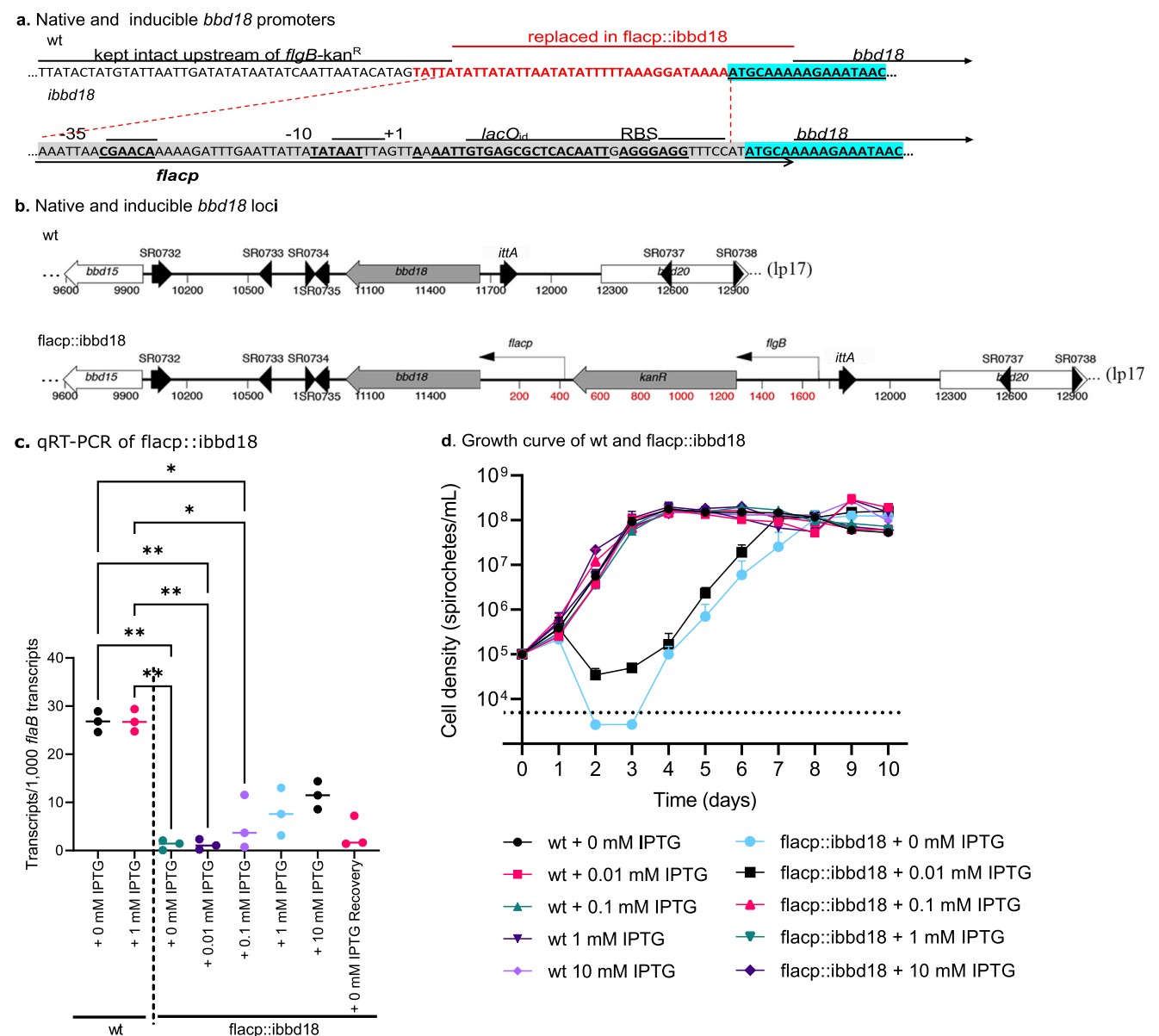

**Fig. 1 | BBD18 is required for B. burgdorferi growth in vitro. a** Native *bbd18* and inducible *ibbd18* upstream sequences. A portion of lp17 (shown in red) was replaced by the IPTG-inducible promoter *flacP*, shown highlighted in grey with the -35 and -10 sigma binding sites, the +1 transcription start site, the LacI binding site (*lacO*id), and the ribosome binding site (RBS). The start of the *bbd18* coding sequence is highlighted in blue. **b** Genetic organization of lp17 in the wt and flacp::ibbd18 strains. Native organization of lp17 around *bbd18* is shown as wt. The genes and annotated small RNAs (SR) are labelled. To create the flacp::ibbd18 strain, a *flgB*-driven kanamycin resistance cassette was placed upstream of the IPTG-inducible *flacP* promoter that drives expression of *bbd18*. **c** qRT-PCR of *bbd18* transcript from wt, flacp::ibbd18 cells, and flacp::ibbd18 recovery cells. Data are presented as individual values of biological triplicates with the mean value indicated. Dunn's multiple comparison of the Kruskal-Wallis test was used to determine significance. **d** Growth curve of the *B. burgdorferi* wt (B31-68-LS) and flacp::ibbd18 strains performed in triplicate with different concentrations of IPTG. Data are presented as the mean values of three biological replicates +/− SD. Cell density was determined by a Petroff-Hauser counting chamber and plating when cell numbers were too low to reliably count under dark-field (shown by dashed line). *n* = 3 biologically independent replicates. *$p$ value < 0.05, **$p$ value < 0.005.

wt. As anticipated, the levels of *bbd18* transcript and BBD18 protein were significantly lower in de-induced flacp::ibbd18 cells relative to wt, and inversely correlated with the levels of *rpoS* transcript and RpoS protein detected by RNA-seq and TMT-MS, respectively (Fig. 3). The mechanism by which BBD18 regulates *rpoS* transcript is currently unknown, but no previously identified regulators of *rpoS*, such as *bb0764*/HK2, *bb0763*/Rrp2, *bb0450*/RpoN, *bb0420*/HK1, *bb0419*/Rrp1, *bb0733*/PlzA, and *bb0647*/BosR, differed significantly at either the RNA or protein level after de-induction of *ibbd18* in flacp::ibbd18 cells (Supplemental Figs. S2 and S3). To further probe the apparent regulator function of BBD18, computational approaches were utilized. Similar to our previous analyses[61], ITASSER predicted structural

similarity of BBD18 to nucleic acid binding proteins[65–69]. Furthermore, an AlphaFold model of BBD18 with a DNA ligand places the previously identified amino acid residues critical for BBD18 regulatory activity at the interface with DNA (Supplemental Fig. S4)[61]. Hence these current structural predictions further implicate BBD18 as a DNA/RNA binding protein that (directly or indirectly) regulates *rpoS*.

Although never attaining a wt level of *bbd18* transcript, the growth phenotype of induced flacp::ibbd18 was indistinguishable from that of wt (Figs. 1 and 3). Therefore we reasoned that genes pertinent to the lytic phenotype of de-induced flap::ibbd18 cells should be differentially expressed relative to both wt and induced flacp::ibbd18 cells. When comparing the transcriptomes of de-induced flacp::ibbd18 cells with wt

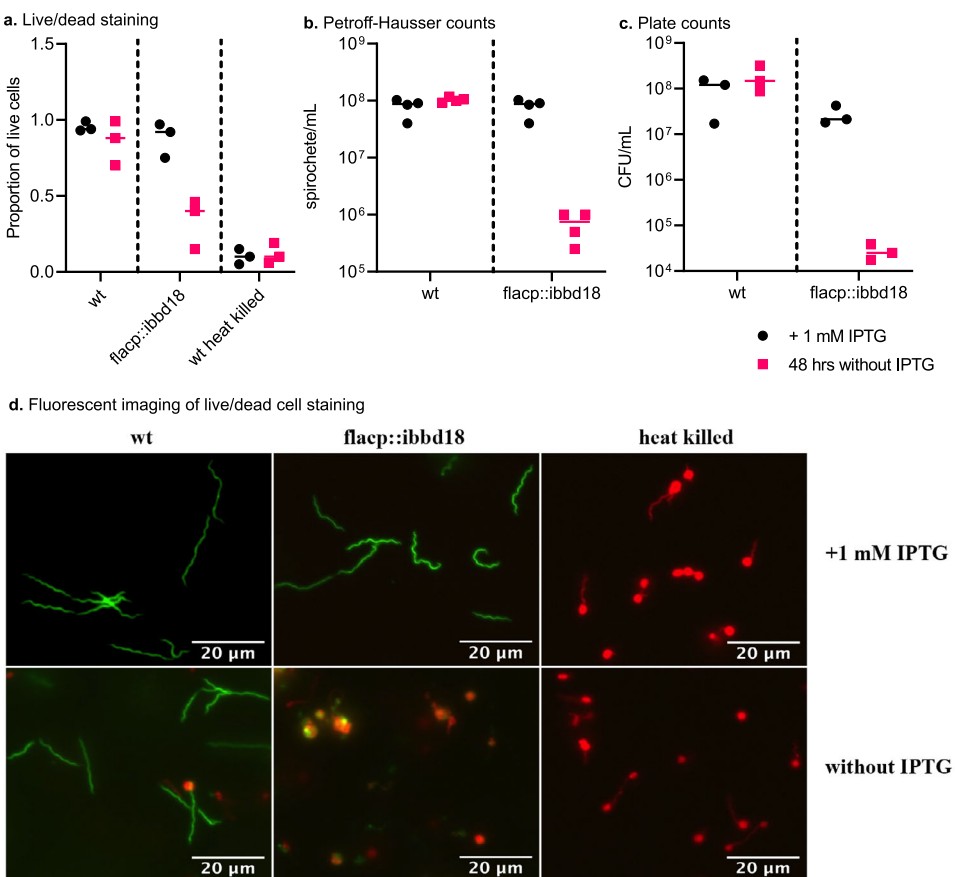

**Fig. 2 | Lysis of flacp::ibbd18 cells. a** Live/dead staining was performed to determine the proportion of live and dead cells; heat-killed wt cells were used as a control. **b** Dark field microscopy was used to determine the number of spirochetes per mL using a Petroff-Hausser counting chamber and **c** cells were plated to determine viable spirochetes as CFU/mL after 48 hrs. with or without IPTG. Data are presented in all graphs as individual values of biological replicates with the mean value indicated. **d** Cells fluorescently labeled with SYTO 9 and propidium iodine were imaged using either a FITC or TRITC channel and then merged to identify live (green) and dead (red) cells 48 hrs after inclusion or omission of 1 mM IPTG; heat-killed wt cells were used as a control. This was repeated with three biological replicates with representative images shown. 2a $n = 3$ biological replicates. 2b $n = 4$ biological replicates. 2c $n = 3$ biological replicates.

(or induced flacp::ibbd18 cells), 313 (301) genes were significantly up-regulated, while 98 (70) genes were significantly down-regulated (Supplementary Data 1: up-regulated and down-regulated & Supplemental Fig. S1). Most of the genes that were differentially expressed in flacp::ibbd18 +/− IPTG induction are also differentially expressed in the comparison with wt (267/301 up-regulated and 54/70 down-regulated) (Supplemental Fig. S1). About a quarter of the up-regulated (83/313) and down-regulated (18/98) genes in de-induced flacp::ibbd18 relative to wt have been previously identified as part of the RpoS-regulon[24,25]. These include *ospC, bba66, bbj24,* and *bba34,* which are carried on plasmids that encode many previously identified RpoS-regulated genes (Supplementary Data 1, Supplemental Fig. S5)[24,25]. While many of the up-regulated genes are encoded in tandem on their respective plasmids, their increased expression does not reflect read-through from surrounding genes (Supplemental Fig. S6). Interestingly, de-induced flacp::ibbd18 cells exhibited increased expression of *bbd20-22,* which are telomeric to *bbd18* on lp17 (Supplementary Data 1: up-regulated). The increased expression of *bbd20-bbd22* preceding cell lysis was reminiscent of our previous demonstration of spirochete survival without BBD18 in the context of a deletion encompassing *bbd15-25* on the right telomeric half of lp17[61]. We also noted that 130 up-regulated genes in de-induced flacp::ibbd18 relative to wt are encoded on the cp32 prophage plasmids, many of which have been identified as RpoS-regulated (Supplementary Data 1: upregulated)[24,25,70,71].

Expression of small RNAs (sRNAs) in *B. burgdorferi* has been investigated in vitro, and one sRNA, SR0736/*ittA,* has been reported to

be involved in tissue tropism, with inactivation leading to significant attenuation of infectivity[72]. Therefore, we assessed the expression of sRNAs and found 118 small RNAs (sRNAs) that were significantly up-regulated in de-induced flacp::ibbd18 cells (Supplementary Data 1: up-regulated). Of these, 51 sRNAs were previously reported to be significantly up-regulated in *B. burgdorferi* cultured at 37 °C, a temperature shift used to induce RpoS-regulated genes in vitro[7,29,73]. Of note, we did not detect expression of sRNA SR0736/*ittA,* encoded between *bbd18* and *bbd20* on lp17, by RNA-seq analysis of all cultures (wt and induced/de-induced flacp::ibbd18). The *ittA* sequence and surrounding DNA were not altered by insertion of the inducible promoter upstream of *bbd18* (Fig. 1). Further investigation by qRT-PCR analysis revealed low levels of *ittA* expression in wt and flacp::ibbd18 spirochetes (Supplementary Data 1 and Supplemental Fig. S7)[72]. While not significantly different, *ittA* levels are slightly elevated in de-induced flacp::ibbd18 cells compared to wt and induced flacp::ibbd18 cells, which corresponds to previous reports of increased *ittA* expression with RpoS induction at 37 °C (Supplemental Fig. S7)[73]. We do not know if a modest difference in genome content accounts for the observed disparity in *ittA* expression, but different B31 wt clones were used in these studies.

**Deletion of *rpoS* rescues the lethal phenotype of flacp::ibbd18**

Over-expression of *bbd18* inhibits RpoS-regulated gene expression[60–62,74], while de-induction of *ibbd18* results in a significant increase in *rpoS* transcript and RpoS protein (Fig. 3 & Supplementary Data 1). Additionally, lethal engineered over-expression of *rpoS* causes

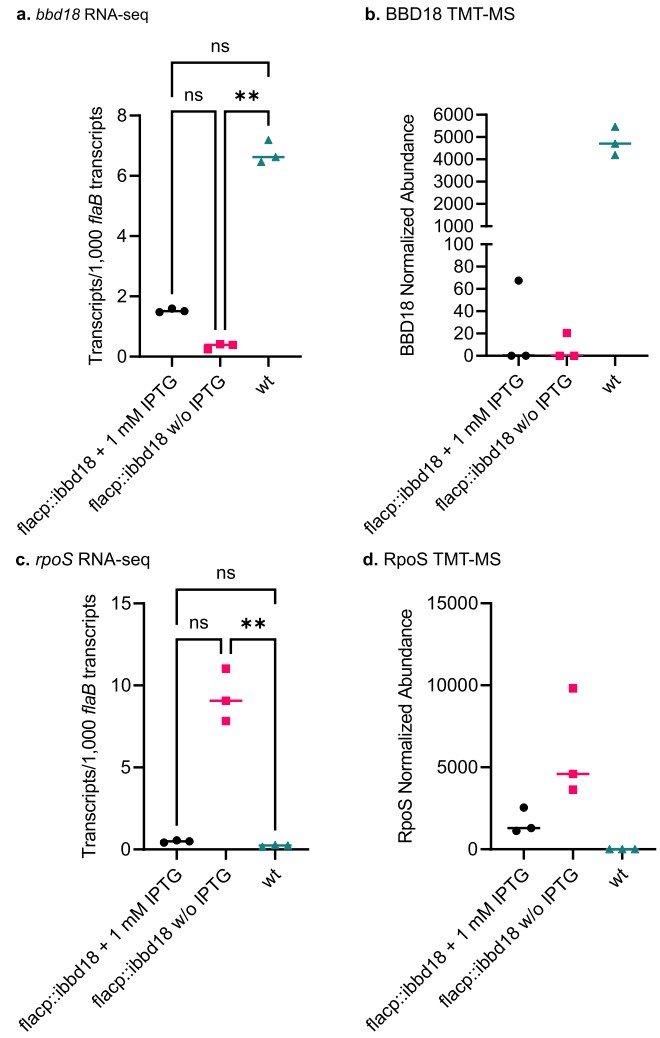

**a. bbd18 RNA-seq**

**b. BBD18 TMT-MS**

**c. rpoS RNA-seq**

**d. RpoS TMT-MS**

**Fig. 3 | Expression of *bbd18*/BBD18 and *rpoS*/RpoS.** RNA-seq (**a/c**) and TMT mass-spec (**b/d**) of *bbd18*/BBD18 and *rpoS*/RpoS. Values were normalized to *flaB*/FlaB. Protein levels mirror transcript levels. Data are presented in all graphs as individual values of biological replicates with the mean value indicated. Dunn's multiple comparison of the Kruskal-Wallis test was used to determine significance. Three biological replicates of all three samples underwent TMT mass-spec analysis; however, RpoS was only detected in flacp::ibbd18 cultures. $n = 3$ biological replicates. *$p$ value < 0.05, **$p$ value < 0.005.

*B. burgdorferi* to bleb and lyse[33], similar to what is seen in de-induced flacp::ibbd18 cells when analyzed by transmission electron microscopy (Fig. 4a–c). Therefore, we constructed isogenic *rpoS* deletion mutants in both wt and flacp::ibbd18 backgrounds (ΔrpoS and flacp::ibbd18ΔrpoS) to determine if cell death following de-induction of *ibbd18* were *rpoS*-dependent (Fig. 4d). As anticipated, deletion of *rpoS* had no impact on growth of wt and induced flacp::ibbd18 cells cultured in media containing 1 mM IPTG (Fig. 4e)[26]. Strikingly, however, deletion of *rpoS* eliminated the dependence of flacp::ibbd18 cells on IPTG-induction for growth (Fig. 4e). RNA-seq analyses confirmed that *bbd18* transcript was absent in flacp::/ibbd18ΔrpoS cells cultured without IPTG (Supplementary Data 1: ΔrpoS down-regulated and Fig. 4f), demonstrating that growth of de-induced flacp::/ibbd18ΔrpoS resulted from deletion of *rpoS* and not from a suppressor mutation that restored *ibbd18* expression without induction. Although de-induction of flacp::ibbd18 led to increased expression of around 300 genes (Supplementary Data 1: up-regulated and Supplemental Fig. S1a), there were no genes with increased expression in de-induced

relative to induced flacp::ibbd18ΔrpoS (Supplementary Data 1: ΔrpoS up-regulated and Supplemental Fig. S1b). This outcome validates our initial interpretation that the ~300 genes with increased expression in de-induced flacp::ibbd18 are a consequence of RpoS-induction and thereby identifies a much larger set of genes comprising the RpoS-dependent regulon than previously recognized. This outcome also confirms that cell death arises as a down-stream consequence of unmodulated RpoS-induction and does not stem directly from the absence of BBD18.

**Scrutinizing ΔrpoS transcriptomes for regulatory intermediates**

In addition to establishing the global transcriptome that follows RpoS induction, we analyzed the RNA-seq datasets from the ΔrpoS strains to identify potential transcriptional regulatory intermediates between BBD18 and RpoS. Simplistically, BBD18 could directly repress transcription of *rpoS* without an intermediate. Less parsimoniously, BBD18 could negatively regulate an activator or positively regulate a repressor of *rpoS* transcription. The expression of these hypothetical intermediates should increase (activator) or decrease (repressor) with de-induction of flacp::ibbd18ΔrpoS cells (decreased BBD18) and corresponding induction of RpoS. Complicating a straightforward interpretation of differential gene expression in a ΔrpoS background is the possibility that BBD18 may regulate genes that do not influence *rpoS* transcription.

With that caveat in mind, we identified a limited number of genes whose expression was positively or negatively influenced by BBD18 in the ΔrpoS background. Of these, 15 genes (all plasmid-encoded) and one sRNA decreased in expression when BBD18 was removed in ΔrpoS spirochetes (de-induced relative to induced flacp::ibbd18ΔrpoS cells) (Supplementary Data 1: ΔrpoS down-regulated and Supplemental Fig. S1b). This sRNA (SR0852) was not previously identified as differentially expressed during RpoS induction, nor was it down-regulated in de-induced flacp::ibbd18 in the current study (Supplementary Data 1: ΔrpoS down-regulated), rendering it an unlikely candidate for a key RpoS-regulatory role. Many (11/15) of the BBD18-dependent genes in the ΔrpoS background were previously identified as down-regulated in RpoS-induced cells and a subset of these (9/11) were also identified as down-regulated in flacp::ibbd18 de-induced cells in the current study (Supplementary Data 1: down-regulated and ΔrpoS down-regulated, Supplemental Fig. S1)[24,25]. The proteins encoded by these and two additional BBD18-dependent genes (*bba68, bba69, bbi36, bbi38, bbi39, bbj08, bbj09 bbj10, bbj41, bbk01, bbk15* and *bbg01-02*) have been identified or annotated as cell envelope proteins, diminishing the likelihood that they serve as transcriptional regulators.

The remaining genes influenced by BBD18 in the ΔrpoS background, *bbd20* and *bbd22*, in addition to the one up-regulated gene *bbd23* (Supplementary Data 1: ΔrpoS up-regulated), are telomeric to *bbd18* on lp17. These genes are annotated as encoding transposon-like proteins with authentic frameshift mutations (*bbd20, bbd23*) and a hypothetical protein of unknown function (*bbd22*). However, decreased expression of *bbd20* and *bbd22* in de-induced /ΔrpoS cells contrasts sharply with their pattern of expression in the presence of RpoS, where de-induction of flacp::ibbd18 results in a significant increase in *bbd20* and *bbd22* transcripts; this pattern suggests modulation of *bbd20* and *bbd22* expression by both BBD18 and RpoS, rather than a potential regulatory role between them (Supplementary Data 1: down-regulated and ΔrpoS down-regulated). Intriguingly, the only gene whose expression increased in flacp::/ibbd18ΔrpoS (with or without induction) relative to ΔrpoS spirochetes was *bbd23*, possibly reflecting the reduced level of BBD18 in the induced strain relative to wt (Supplementary Data 1: ΔrpoS up-regulated). However, unlike *bbd20* and *bbd22*, expression of *bbd23* did not increase relative to wt with de-induction of flacp::ibbd18 (Supplementary Data 1: upregulated), again incompatible with a direct RpoS-modulatory role. Whatever role(s) these lp17 genes

**a.** TEM of flacp::ibbd18 + 1 mM IPTG

**b.** TEM of flacp::ibbd18 without IPTG for 24 hrs

**c.** TEM of flacp::ibbd18 without IPTG for 40 hrs

**d.** ΔrpoS construct

**e.** Growth curve of strains

- flacp::ibbd18 + 1 mM IPTG
- flacp::ibbd18 w/o IPTG
- flacp::ibbd18ΔrpoS + 1 mM IPTG
- flacp::ibbd18 ΔrpoS w/o IPTG
- wt + 1 mM IPTG
- wt w/o IPTG
- ΔrpoS + 1 mM IPTG
- ΔrpoS w/o IPTG

**f.** *bbd18* RNA-seq

1. flacp::ibbd18ΔrpoS + 1 mM IPTG

2. flacp::ibbd18ΔrpoS w/o IPTG

3. ΔrpoS w/o IPTG

**Fig. 4 | Unmodulated expression of *rpoS* following de-induction of *bbd18* is responsible for *B. burgdorferi* lysis in vitro.** (**a-c**) Representative TEM images of flacp::ibbd18 cells cultured with or without IPTG. **a** flacp::ibbd18 cells are intact in the presence of IPTG/BBD18. Flacp::ibbd18 cells bleb and lyse (**b** and **c**) after the removal of IPTG from the culture medium. TEM was performed on spirochetes from two biological replicates with representative images shown. **d** Genetic organization of the *rpoS* locus in wt and ΔrpoS strains. **e** Growth curve of *B. burgdorferi* flacp::ibbd18, flacp:: ibbd18ΔrpoS, wt, and ΔrpoS strains in the absence or presence of 1 mM IPTG; error bars show the standard deviation. Data are presented as the mean values of three biological replicates +/− SD. **f** *bbd18* transcripts are not restored in flacp::ibbd18ΔrpoS cells cultured in the absence of IPTG, demonstrating that the ability of these cells to grow without IPTG induction is due to the deletion of *rpoS* and not to a mutation rescuing expression of *bbd18*. Data are presented in all graphs as individual values of biological replicates with the mean value indicated. Dunn's multiple comparison of the Kruskal-Wallis test was used to determine significance. 4e-f *n* = 3 biological replicates. *p-value < 0.05.

play in wt cells, their differential expression does not result in cell death when BBD18 is removed in a /ΔrpoS background.

Finally, expression of yet another gene on lp17, *bbd21*, increased following de-induction of flacp::ibbd18, but this did not occur in the /ΔrpoS strains, suggesting that *bbd21* expression is RpoS-dependent (Supplementary Data 1: upregulated, ΔrpoS up-regulated and ΔrpoS down-regulated). Therefore, without clear evidence to the contrary, we will retain a parsimonious model in which BBD18 directly represses *rpoS* transcription and thereby indirectly regulates the entire RpoS regulon.

## Depletion of *bbd18* leads to prophage induction

The observed RpoS-dependent increase in cp32 prophage gene expression[24,25,42–44] (Supplementary Data 1: up-regulated) led us to hypothesize that prophages were induced when RpoS was over-expressed in the absence of BBD18. To this end, we evaluated plasmid copy number and determined that only the cp32 and lp28-2 prophage plasmids significantly increased relative to the copy number of the chromosome in flacp::ibbd18 cells following removal of IPTG; although not significant, an increase in copy number of lp28-1 was also evident (Fig. 5a, b). Additionally, qRT-PCR of RNA extracted from flacp::ibbd18

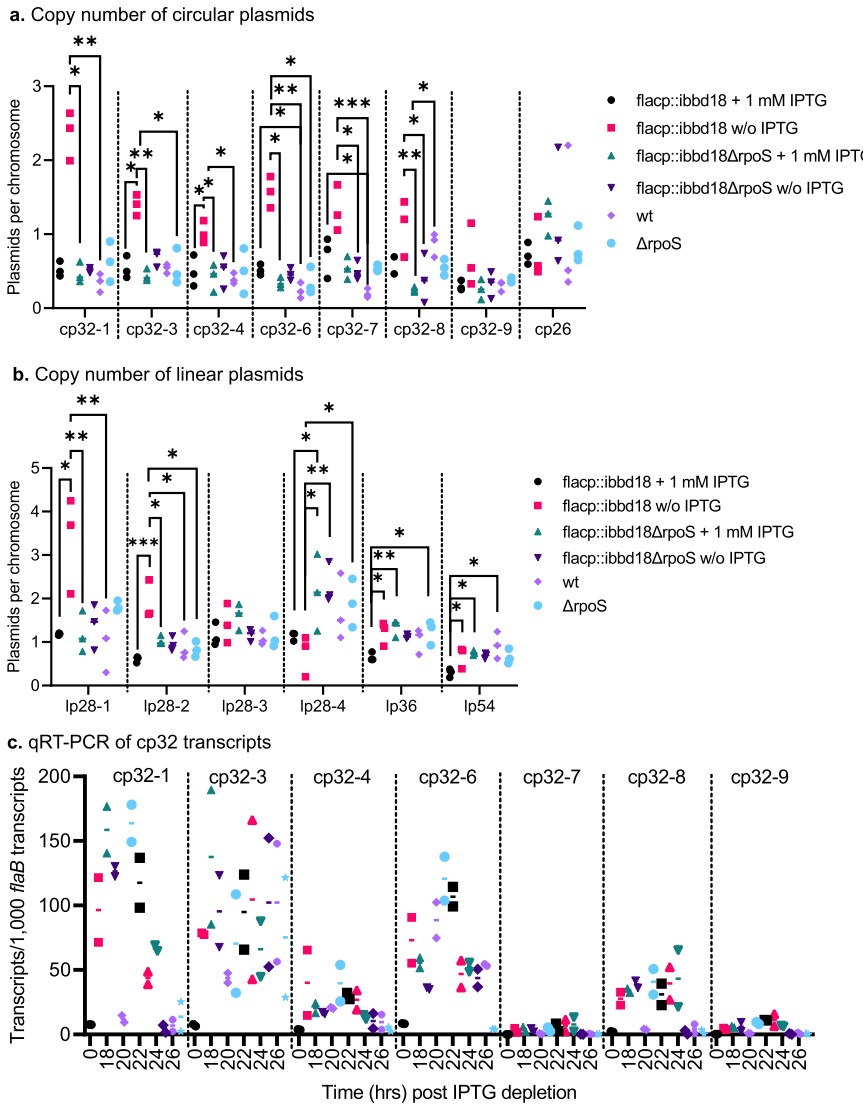

**Fig. 5 | Depletion of BBD18 leads to increased replication and transcription of prophage plasmids. a, b** Copy number of linear and circular plasmids. The cp32 and lp28-2 prophage plasmid copy numbers significantly increased only in flacp::ibbd18 when cultured for 40 hrs in the absence of IPTG and analyzed with Dunn's multiple comparison of the Kruskal-Wallis test. **c** Transcripts of the cp32-1, cp32-3, cp32-4, cp32-6, and cp32-8 plasmids increased in flacp::ibbd18 following IPTG depletion of a culture started at $10^6$ spirochetes per ml. Data are presented in all graphs as individual values of biological replicates with the mean value indicated. 5a-b $n = 3$ biological replicates. 5c $n = 2$ biological replicates. *$p$ value < 0.05, **$p$ value < 0.005, ***$p$ value < 0.0005.

cells at various time points following removal of IPTG demonstrated increased expression of representative phage genes from cp32-1, cp32-3, cp32-6, and cp32-8 plasmids following de-induction of flacp::ibbd18 (Fig. 5c). RNA-seq analysis also identified a significant increase in expression of the majority of cp32 prophage genes in the absence of *bbd18* (Supplementary Data 1: up-regulated). Most of the cp32-encoded proteins that could be detected with TMT-MS analysis were more abundant in de-induced flacp::ibbd18 (Supplementary Data 1: TMT-MS). The relative absence of cp32 proteins detected by TMT-MS may be explained by covalent linkage of phage capsid proteins causing resistance to trypsin digestion.

**Phage in culture supernatants of de-induced flacp::ibbd18**
The increase in cp32 plasmid copy number and transcripts (Fig. 5) prompted us to investigate the presence of phage virions in flacp::ibbd18 cultures. Cell-free supernatants of flacp::ibbd18 and wt cultures grown in the presence or absence of IPTG were subjected to DNA extraction and electron microscopic imaging. Agarose gel

electrophoresis of DNA purified from sterile culture supernatants treated with DNase prior to phenol extraction revealed a discrete band of DNA, approximately midway between the 50 kb and 14 kb size standards, from the supernatants of de-induced flacp::ibbd18 cultures, but not from induced flacp::ibbd18 or wt culture supernatants (Fig. 6a). The lack of clearly resolved markers in this region of the gel prevent an accurate estimation of size. However, this DNA species, shielded from DNase, is compatible in size with the genomes of the endogenous lp28-2 or cp32 prophage. Furthermore, transmission electron microscopic images revealed the presence of two distinct phage morphologies in the supernatant of de-induced flacp::ibbd18 cultures (Fig. 6b), similar to what was previously observed by Nuebert et al. following ciprofloxacin treatment of a clinical *B. burgdorferi* isolate[48]. One of these phages resembles the well-characterized cp32 ΦBB-1 phage with a polyhedral head described by Eggers et al.[75]. These data demonstrate that endogenous prophages are induced in response to unmodulated *rpoS* expression in de-induced flacp::ibbd18 cells.

## Lysis of flacp::ibbd18 cells lacking cp32s

The increase in copy number of cp32 prophage plasmids relative to the chromosome, the corresponding increase in cp32 prophage gene expression, the detection of phage particles, and the detection of cp32 DNA in DNase-treated supernatant of de-induced flacp::ibbd18 but not flacp::/ibbd18ΔrpoS cultures (Supplemental Fig. S8a), prompted us to speculate that lysis of spirochetes lacking BBD18 was due to cp32 prophage induction. To test this hypothesis, we displaced the entire set of cp32 plasmids from both wt and flacp::ibbd18 spirochetes. This was accomplished through sequential rounds of transformation with 7 distinct shuttle vectors (SV). Each SV contained a fragment of B31 cp32 DNA encompassing a partial set of plasmid replication and partition genes, extending from *nucP* to the PFam 32 gene, from a single cp32 plasmid (e.g. *bbp29-bbp32* of cp32-1) (Fig. 7a). These replication-competent SVs were incompatible with their cognate plasmid, thus displacing the cp32 from which they were derived. However, they were unstable due to omission of the *parB*/PFam 49 gene and readily lost when antibiotic selection was lifted. After a clonal transformant was confirmed to lack both the targeted cp32 and the displacing SV, it was subjected to another round of transformation with a different SV targeting a remaining cp32. This process was repeated until all 7 cp32 plasmids had been sequentially displaced from both wt and flacp::ibbd18 cells (Fig. 7b).

No growth defect was detected in either wt or flacp::ibbd18 cells lacking all seven cp32 plasmids when grown with IPTG induction (Fig. 7c). However, removal of IPTG still resulted in impaired growth and lysis of flacp::ibbd18 cells, even those lacking cp32 plasmids, albeit somewhat delayed (Fig. 7c). Therefore, while cp32 prophage are not solely responsible for the lytic phenotype of de-induced flacp::ibbd18, they do impact cell viability. Additionally, as two different phage types were detected in the supernatants of de-induced flacp::ibbd18 cultures, an additional unrelated endogenous prophage, such as lp28-2, may also contribute to lysis. Indeed, induction of lp28-2 phage is suggested by increased plasmid copy number data and increased lp28-2 gene expression in de-induced flacp::ibbd18 cells (Fig. 5b and Supplementary Data 1: up-regulated), and the increased amount of lp28-2 DNA in the supernatant of de-induced flacp::ibbd18 cells lacking all seven cp32 plasmids (Supplemental Fig. S8b). The level of lp28-2 DNA detected in the supernatant of flacp::ibbd18 cells remains consistent between induced and de-indced flacp::ibbd18 cells, whereas it is significantly increased in de-induced cells when all seven cp32 prophage plasmids are absent. In fact, de-induced flacp::/ibbd18Δcp32 cells have a level of supernatant lp28-2 similar to cp32 DNA in de-induced flacp::ibbd18 supernatants (Supplemental Fig. S8b). However, efforts to displace the lp28-2 plasmid from wt and flacp::ibbd18 spirochetes have not succeeded to date and therefore the contribution of lp28-2 prophage to cell lysis, while strongly suspected, remains unconfirmed.

## Potential role of BBD21/ParA in prophage induction

As described above, expression of the lp17 gene *bbd21* increased in de-induced flacp::ibbd18 spirochetes relative to induced or wt cells, but was not differentially expressed in ΔrpoS cells, consistent with RpoS-dependent expression (Supplementary Data 1: up-regulated, drpoS up-regulated and drpoS down-regulated). Additionally, as mentioned above, we have also previously demonstrated that *bbd18* deletion can be tolerated by non-infectious spirochetes that are unable to undergo RpoS-induction and by infectious spirochetes that lack the right telomeric portion of lp17, including *bbd21*[61]. We therefore hypothesized that BBD21, a PF-32/ParA homolog required for plasmid replication, might be responsible for increased lp28-2 and cp32 prophage plasmid copy number and induction of lytic phage in de-induced flacp::ibbd18 cells. However, deletion of *bbd21* did not rescue the requirement for BBD18 in flacp::ibbd18 cells, indicating that if involved, *bbd21* is not the sole determinant on lp17 through which RpoS regulates

phage lysis-lysogeny decisions (Supplemental Fig. S9 and Wachter, unpublished data).

## Specific role for BBD18 in mouse-tick infectious cycle

Expression of *bbd18* is required for growth of *B. burgdorferi* in culture; therefore, we hypothesized that expression of *bbd18* would also be essential in vivo. To test this hypothesis, wt and flacp::ibbd18 spirochetes were needle-inoculated into mice, where IPTG induction is not maintained. Contrary to our expectations, flacp::ibbd18 spirochetes were able to persistently infect mice, as confirmed by positive serological responses at three weeks post-feeding and recovery of spirochetes from tissues at five weeks after inoculation (all 7 mice inoculated with flacp::ibbd18 spirochetes became infected) (Supplementary Data 1: animal infections). After leaving the IPTG-free in vivo environment, flacp::ibbd18 cells isolated from mouse tissues still required IPTG for in vitro growth, demonstrating that flacp::ibbd18 remained IPTG-inducible despite persisting in an uninduced state in vivo (Supplemental Fig. S10a). The spirochete loads in mouse tissues were also analyzed to determine if in vivo replication and spirochete burden were lower in mice infected with the flacp::ibbd18 strain. Unexpectedly, qPCR of genomic DNA extracted from infected mouse ears identified a significant increase in spirochete burden in flacp::ibbd18-infected relative to wt-infected mice (Fig. 8a).

**a.** Protected DNA in culture supernatants

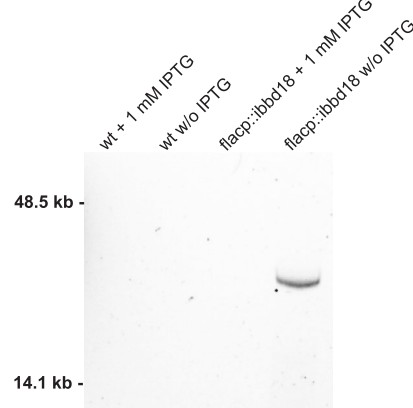

**b.** TEM of phage isolated from flacp::ibbd18

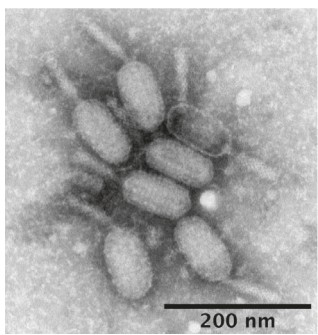 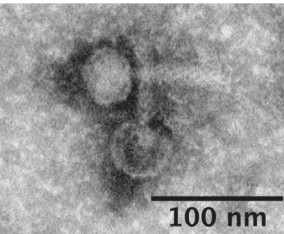

**Fig. 6 | Presence of phage in the cell-free supernatant of BBD18-depleted cells.** **a** A discrete band of DNA was detected in the DNase-treated supernatant of flacp::ibbd18 cells cultured in the absence of IPTG. **b** Supernatant of flacp::ibbd18 cells cultured in the absence of IPTG contains at least two types of phage particles, similar to what was previously observed by Neubert et al. following ciprofloxacin treatment of a clinical *B. burgdorferi* isolate[48]. One of the detected phage types contains a polyhedral head akin to the well characterized cp32 ΦBB-1 phage described by Eggers et al.[75]. DNA extraction and TEM were performed on supernatant purified from two separate biological replicates, one for DNA extraction and one for TEM.

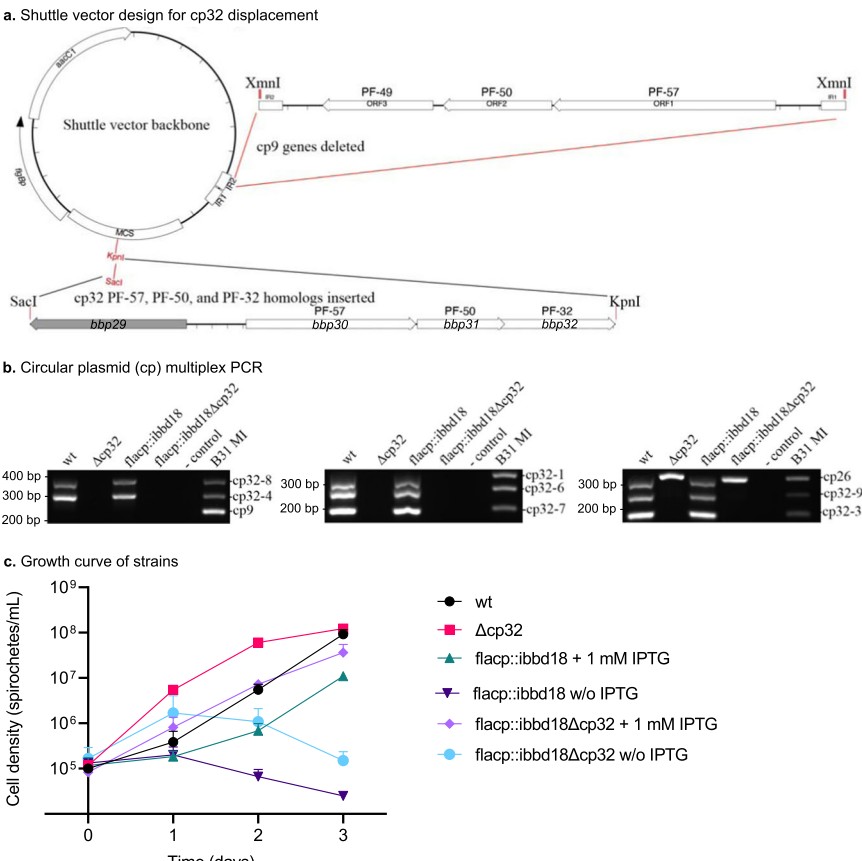

**Fig. 7 | The cp32 plasmids are not solely responsible for cell lysis in the absence of BBD18. a** Shuttle vector design to displace cp32 plasmids with incompatible, but unstable plasmids. PF-57, PF-50, and PF-32 support shuttle vector replication and are incompatible with their cognate plasmids, but are inherently unstable because the PFam49/*parB* gene is not included. **b** Multiplex PCR of circular plasmids in isogenic B31 wt (A3-68-LS), △cp32, flacp::ibbd18, and flacp::ibbd18△cp32 strains. B31 MI is a positive control containing all plasmids. The agarose gel presented is a representative image of five multiplex PCRs performed at different dates. **c** Growth curve of B31 wt (A3-68-LS), △cp32, flacp::ibbd18, and flacp::ibbd18△cp32. Data are presented as the mean values of three biological replicates +/− SD. Cell numbers were monitored daily by dark-field microscopy using a Petroff-Hauser counting chamber and plated to quantify viable bacteria as CFU when cell numbers were too low to reliably count under dark-field. *n* = 3 biological replicates.

To evaluate the ability of spirochetes to infect and survive in the tick vector, larvae were fed to repletion on wt- and flacp::ibbd18-infected mice and spirochete burdens were determined. When assessed at 1 week, 7 weeks, and ~16 weeks post repletion, spirochetes were recovered from 100% (13/13), 70% (7/10), and ~82% (9/11) of larvae fed on wt-infected mice, and from ~62% (8/13), 90% (9/10), and ~82% (9/11) of larvae fed on flacp::ibbd18-infected mice, respectively (Supplementary Data 1: animal infection). The spirochete burdens in unfed nymphs were also similar between *Ixodes* infected with wt or flacp::ibbd18 strains (Fig. 8b). All flacp::ibbd18 spirochetes recovered from *Ixodes* required the addition of IPTG to the growth medium, demonstrating that the flacp::ibbd18 strain retained the in vitro requirement for IPTG and that *bbd18* gene expression is not required for larval tick acquisition and transstadial transmission through the molt to the nymphal stage (Supplemental Fig. S10b).

The retention of the flacp::ibbd18 strain by unfed nymphs provided an opportunity to assess the ability of flacp::ibbd18 to infect mice by tick bite, the natural mode of transmission for mammalian infection. Therefore, wt- and flacp::ibbd18-infected flat nymphs were fed on naïve mice and the number of spirochetes they carried was assessed during feeding (24 hrs after attachment), after feeding to repletion, and subsequently at 24 hrs, 48 hrs, 10 days, and 1 month after drop-off. Similar to challenge by needle inoculation, wt and flacp::ibbd18 spirochetes were infectious for mice by tick bite, with the majority of mice seropositive at three weeks and spirochetes isolated from mouse tissues at 5 weeks post-infection (Supplementary Data 1: animal infections). While spirochete loads were similar in unfed wt- and flacp::ibbd18-infected nymphs, they diverged dramatically following feeding (Fig. 8b). Whereas wt spirochetes increased in number after feeding, peaking at approximately 10^7 spirochetes per nymph, the number of flacp::ibbd18 spirochetes decreased sharply in fed ticks, with no viable spirochetes detected by one month after the nymphal blood meal (Fig. 8b), demonstrating that BBD18 is required for spirochete persistence in the tick midgut after the nymphal blood meal.

Pointedly, flacp::ibbd18 spirochetes isolated from infected mice and nymphs maintained their dependence upon IPTG-induction of *ibbd18* for in vitro growth, indicating that they were not escape mutants (Supplementary Data 1: animal infections, Supplemental Fig. S10). To determine if over-expression of *rpoS* in vivo corresponded with lysis of flacp::ibbd18 spirochetes in fed ticks, qRT-PCR was performed on RNA purified from infected nymphs. This analysis revealed significantly increased *rpoS* expression in flacp::ibbd18 compared to wt spirochetes in infected nymphs that had fed to repletion (Fig. 8c), which was also the only point of the mouse-tick infectious cycle at which *rpoS* transcript could be detected in either wt or flacp::ibbd18 spirochetes. Consistent with this finding, OspC, a hallmark of *rpoS*-dependent gene expression, was not present on wt or flacp::ibbd18 spirochetes in unfed infected nymphs but was present in fed nymphs, as assessed by immuno-fluorescent staining (Supplemental Fig. S11a–d). Although OspA was not detected by IFA on

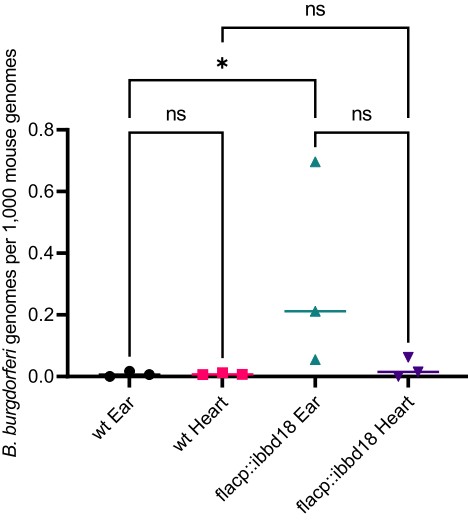

**a.** Spirochete burden in infected tissues

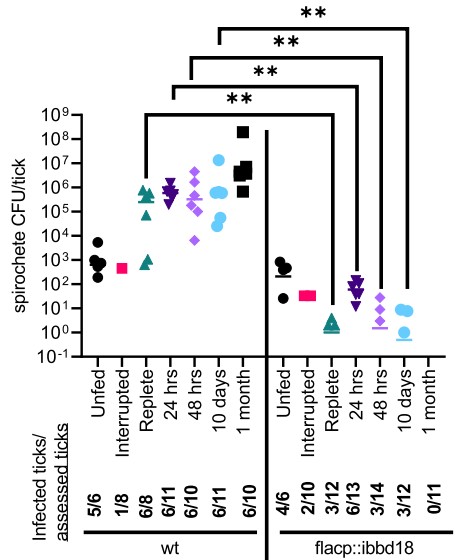

**b.** Viable spirochetes in nymphs

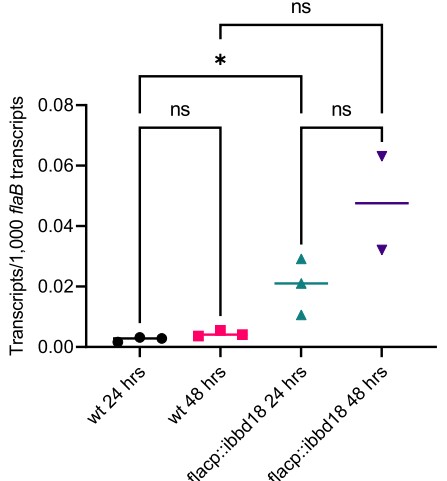

**c.** *rpoS* expression in fed nymphs

**d.** Retention of ΔrpoS spirochetes in ticks

| *Ixodes* life stage | Number of infected ticks/Number of ticks assessed | |
|---|---|---|
| | ΔrpoS | flacp::ibbd18ΔrpoS |
| **Replete larvae** | 1/5 | 3/5 |
| **Unfed nymphs** | 2/2 | 2/2 |
| **Fed nymphs 24 hrs after drop-off** | 1/4 | 3/4 |
| **Fed nymphs 48 hrs after drop-off** | 2/5 | 1/5 |
| **Fed nymphs 10 days after drop-off** | 0/5 | 2/4 |
| **Unfed adults** | 3/8 | 5/5 |

**Fig. 8 | flacp:ibbd18 spirochetes in infected mice and ticks. a** qPCR of gDNA extracted from ears of mice after 5 weeks of infection revealed a higher burden in ears of flacp::ibbd18 than wt spirochetes. Dunn's multiple comparison of the Kruskal-Wallis test was used to determine significance. **b** Spirochete loads in *I. scapularis* nymphs prior to and following feeding as assessed by plating. Mann-Whitney test found significant differences between wt and flacp::ibbd18 loads at every point for which viable flacp::ibbd18 spirochetes were detected in infected nymphs following feeding (replete, 24 hrs, 48 hrs, and 10 days); no flacp::ibbd18 spirochetes were recovered from ticks at 1 month after drop-off. The Mann Whitney nonparametric test was used to determine significance. **c** qRT-PCR

of total RNA extracted from fed nymphs revealed a significant increase in the level of flacp::ibbd18Δ*rpoS* transcript relative to wt at 24 hrs after drop-off. Dunn's multiple comparison of the Kruskal-Wallis test was used to determine significance. Data are presented in all graphs as individual values of biological replicates with the mean value indicated. **d** Ticks artificially-infected with flacp::ibbd18Δ*rpoS* as larvae retain spirochetes throughout all life stages. Viable flacp::ibbd18Δ*rpoS* spirochetes persisted in fed nymphs through the molt to adult. 8a *n* = 3 biological replicates. 8b *n* = 6 biological replicates. 8c *n* = 3 biological replicates. *p value < 0.05, **p value < 0.005.

flacp::ibbd18 spirochetes in the midguts of fed nymphs, these OspC+ spirochetes also produced a similar level of *ospA* transcript compared to wt cells (Supplemental Fig. S11e, f). These results demonstrate that induction of *rpoS* in flacp::ibbd18 spirochetes in vivo occurs during the nymphal tick blood meal, at the same point as *rpoS* induction in wt spirochetes. Furthermore, deletion of *rpoS* prevented lysis of flacp::ibbd18 spirochetes in vivo, as flacp::ibbd18Δ*rpoS* spirochetes could be isolated from infected nymphs (artificially infected as larvae) at all time points following nymphal feeding (Fig. 8d). Together these results demonstrate that BBD18 functions as a critical modulator of *rpoS* induction in vivo to prevent phage induction and spirochete lysis during and following the nymphal blood meal.

## Discussion

*B. burgdorferi* is an obligate symbiont that exists in disparate environments where it must tightly regulate the expression of genes required for tick acquisition and mammalian infection. Previous work has outlined the role of BBD18 as a negative regulator of RpoS-dependent gene expression and demonstrated that engineered constitutive expression of *bbd18* in wild-type *B. burgdorferi* blocks RpoS induction and prevents host infection[61,62]. However, previous attempts to obtain a *bbd18* deletion mutant in a wild-type background were unsuccessful and the role of *bbd18* had not been fully characterized. In this study we generated a conditional *bbd18* mutant, termed flacp::ibbd18, in a fully infectious wild-type background and defined the role of *bbd18* both in vitro and in vivo.

BBD18 is required for spirochete growth in culture, with de-induction of *ibbd18* leading to unmodulated expression of *rpoS* and cell lysis (Figs. 1–4). Cell death as a consequence of *rpoS* over-expression was previously reported, but the basis for lethality was not determined[33]. RNA-seq and TMT mass-spec of flacp::ibbd18 cells revealed differential regulation of over 300 genes and 118 sRNAs when flacp::ibbd18 was de-induced (Supplementary Data 1: up-regulated and down-regulated, Supplemental Fig. S1). However, the vast majority of these genes are RpoS-regulated, with 0 genes upregulated, and only 15 plasmid genes and 1 sRNA down-regulated when *ibbd18* was de-induced in cells lacking *rpoS* (Supplementary Data 1: /ΔrpoS down-regulated & Supplemental Fig. S1). Eleven of these genes with decreased expression in cells lacking both *ibbd18* expression and *rpoS* encode membrane proteins, one is a pseudogene, and the two remaining genes, *bbg01* and *bbg02*, have no annotated functions, but are predicted to be membrane or phage proteins. The function and expression of the majority of these genes have been investigated and while some have no known role during the infectious cycle (*bbj08, bbj09,* and *bbj41*)[76–78] others are found to be up-regulated under conditions mimicking the tick (*bbk01*)[76,79] or murine (*bba68, bba69, bbg01, bbh37, bbk15, bbi36, bbi38,* and *bbi39*)[76,80–82] environments. Since the majority of regulated genes in flacp::/ibbd18ΔrpoS cells are membrane proteins, BBD18 regulation of *rpoS* appears to be either direct, or indirect through an unidentified intermediary, as known regulators of *rpoS* were not differentially expressed in flacp::ibbd18 cells (Supplemental Figs. S2, S3).

We remain puzzled by the potential contribution of other lp17 genes or sRNAs to the phenotype of spirochetes lacking BBD18. We previously observed that spirochetes retaining *rpoS* could tolerate loss of *bbd18* when it occurred in the context of a larger deletion that removed the right half of lp17[61]. This portion of lp17 is strictly conserved among diverse Lyme disease spirochetes and includes *bbd21,* the primordial PFam 32/ParA gene with presumed plasmid maintenance functions and orthologs on all *Borrelia* plasmids[40,83]. Our interest focused on *bbd21* when we noted elevated expression of *bbd21* after IPTG-induction of *ibbd18* was removed, suggesting a potential link between the ParA protein encoded by *bbd21* and increased plasmid replication/prophage induction in spirochetes lacking BBD18 (Supplementary Data 1: up-regulated). However, deletion of *bbd21* did not circumvent the in vitro growth requirement of spirochetes for BBD18 (Supplemental Fig. S9 and Wachter, unpublished data), indicating that *bbd21* is not the sole driver of RpoS-dependent cell death in vitro. Wong et al. recently reported variation in lp17 copy number in a *B. burgdorferi* mutant lacking *bbd21,* but no profound impact on spirochete growth or infectivity[84]. Hence the identification and contribution of additional lp17 genes to the in vitro and in vivo growth defects of spirochetes lacking BBD18 remain unresolved.

Previous bioinformatics analyses predicted BBD18 to encode a domain with homology to a helix-turn-helix motif (HTH) of SopB, a DNA-binding protein[61]. Point mutations in this predicted DNA-binding region eliminated BBD18's activity as a negative regulator of *rpoS*[61]. Our current ITASSER analyses predicted structural similarity of BBD18 to DNA/RNA binding proteins[65]. An AlphaFold model of BBD18 with a DNA ligand places the previously identified amino acid residues critical for BBD18 regulatory activity at the interface with DNA (Supplementary Fig. S4)[65–69]. Therefore, we suggest that BBD18 is a DNA/RNA binding protein that (directly or indirectly) regulates *rpoS* expression at the transcriptional level, in addition to the post-transcriptional mechanism previously described by Dulebohn et al.[62].

Despite the absolute requirement for BBD18 during in vitro growth, uninduced flacp::ibbd18 spirochetes were infectious in mice. Furthermore, bacteria isolated from mouse tissues maintained their requirement for IPTG-induction/BBD18 in vitro, confirming that they were not escape mutants that lacked a functional *lac* repressor. This demonstration that *B. burgdorferi* does not require BBD18 during murine infection is consistent with our previous results[61] and indicates that *rpoS* expression or RpoS action is modulated by a different mechanism in the vertebrate host following transmission of spirochetes by feeding nymphs. Larval ticks acquired flacp::ibbd18 spirochetes from infected mice, retained them through the nymphal molt, and transmitted flacp::ibbd18 spirochetes to naïve mice (Supplementary Data 1: Animal infections). However, after feeding to repletion, the number of viable spirochetes in flacp::ibbd18-infected nymphs dropped sharply and continued to diminish, until viable spirochetes could no longer be detected by one month after drop-off (Fig. 8b). This is in stark contrast to the dramatic increase in the number of viable spirochetes in wt-infected nymphs during the same time frame (Fig. 8b). This outcome demonstrates that BBD18 is required by *B. burgdorferi* to persist in the tick vector following the nymphal blood meal, when *rpoS* expression would typically be downregulated. RpoS is induced in spirochetes that have colonized the nymphal tick midgut in response to environmental changes that accompany tick attachment and feeding. We hypothesize that BBD18 modulates *rpoS* induction uniquely at this point of the infectious cycle to avoid spirochete lysis. The significant increase in *rpoS* transcript in replete flacp::ibbd18-infected nymphs lacking BBD18 supports this hypothesis (Fig. 8c). Additionally, the continued viability of flacp::ibbd18ΔrpoS spirochetes, which lack both BBD18 and RpoS, demonstrates that lysis of flacp::ibbd18 cells after nymphal feeding is RpoS-dependent (Fig. 8).

The RpoS-dependent expression of genes on the endogenous cp32 prophage plasmids led us to determine that cp32 plasmid copy number and transcripts increase when *ibbd18* induction was removed. Additionally, we identified the presence of a single ~30 kb species of DNase-protected DNA and phage particles in the supernatant of de-induced flacp::ibbd18 cells (Figs. 5, 6). Together, this data supports our hypothesis that the absence of BBD18 results in induction of lytic phage due to RpoS over-expression[24,25,42–44] (Figs. 5, 6). However, displacement of all seven cp32 prophage plasmids delayed, but did not prevent, cell lysis in the absence of *ibbd18* expression (Fig. 7). Consistent with RpoS-dependent induction of at least 2 different phage types (Fig. 6), de-repression of *rpoS* also resulted in increased copy number, gene expression, and the presence of protected prophage lp28-2 DNA, which is unrelated to the cp32 prophage, in the supernatant of flacp::ibbd18Δcp32 cells (Figs. 5, 6, Supplementary Data 1: up-regulated and down-regulated, and Supplemental Fig. S8). Therefore, while induction of the cp32 prophage is not solely responsible for cell lysis in the absence of *ibbd18,* induction of additional endogenous phage, such as lp28-2, may also contribute to cell lysis.

Prophage induction represents an unrecognized component of the RpoS regulon of *B. burgdorferi* during the infectious cycle, particularly in the tick midgut where horizontal gene transfer among heterogenous co-infecting spirochetes could occur[58,85]. Our previous analysis demonstrated that *ospC,* which encodes a highly variable, immunodominant surface protein required for host infection[18,49,52,86], contains a significantly higher number of *Borrelia* restriction-modification (R/M) motifs than surrounding sequences[87]. These R/M motifs could provide breakpoints in the cp26 DNA to facilitate packaging into phage heads and recombination at the native locus. Additionally, there is increased expression of R/M genes in the tick midgut following the nymphal blood meal[87]. Interestingly, this is also the only point during the in vivo mouse-tick infectious cycle at which the phenotypes of flacp::ibbd18 and wt spirochetes diverge. The inferred presence of R/M enzymes and phage in spirochetes colonizing the nymphal tick midgut provide a likely stage for horizontal gene transfer. Previous work by Eggers and colleagues has shown that the cp32 phage, termed ΦBB-1, can transduce cp32 and non-cp32 DNA[45,46]. Transduction of *ospC* sequences is substantiated by detection of cp26 DNA in ΦBB-1 virions (Secor and Kinnersley, unpublished data). We hypothesize that BBD18-modulated, RpoS-dependent induction of

transducing phage in *B. burgdorferi* during tick feeding gives rise to antigenic diversity required for superinfection of immune hosts in local endemic regions[51,54,59,88–90].

The mechanism of cell lysis accompanying *rpoS* over-expression remains undefined. Although unmodulated induction of lytic phage seems a likely explanation, lethality could also arise from competition by excess RpoS for a limited pool of core RNA polymerase, resulting in inadequate expression of essential house-keeping genes[34]. RNA-sequencing of *bbd18*-depleted, RpoS over-expressing cells revealed over 300 differentially regulated genes and 118 sRNAs relative to wt or flacp::ibbd18 cells cultured in the presence of IPTG (Supplementary Data 1: up-regulated and down-regulated, Supplemental Fig. S1). Although expression of most of these genes increased with RpoS-induction, transcript levels for some genes decreased following *rpoS* overexpression. These down-regulated genes include *bb0622* (*ackA*) and *bb0250* (*dedA*), which are required for growth of *B. burgdorferi* in culture and whose decreased expression could contribute to cell death[91,92]. Conversely, RpoS overexpression in the absence of BBD18 is not lethal for spirochetes infecting the murine host. It is presumed that other mechanisms of *rpoS* and RpoS regulation are occurring in the mammalian host, but transcript levels of *rpoS* and known regulators of *rpoS* were undetectable in RNA isolated from infected mouse tissues.

In conclusion, in this study we demonstrate a link between phage induction and the RpoS-dependent adaptive response that Lyme disease spirochetes undergo during tick feeding. Ongoing and future studies are needed to define the mechanism of spirochete death upon de-repression of *rpoS*, the participation of phage in horizontal gene transfer, and the in vivo phenotype of spirochetes lacking all cp32 prophage. We infer that the multipartite structure of the *Borrelia* genome facilitates genetic exchange and reassortment, and thereby fosters diversity of spirochetes in the natural enzootic cycle.

## Methods

### Ethics statement
All animal work was performed according to the guidelines of the National Institutes of Health, *Public Health Service Policy on Humane Care and Use of Laboratory Animals*[93], and the United States Institute of Laboratory Animal Resources, National Research Council, *Guide for the Care and Use of Laboratory Animals*[94]. Protocols were approved by the Rocky Mountain Laboratories, NIAID, NIH Animal Care and Use Committee. The Rocky Mountain Laboratories are accredited by the International Association for Assessment and Accreditation of Laboratory Animal Care (AAALAC). All efforts to minimize animal suffering were made.

### *B. burgdorferi* strains and growth conditions
*B. burgdorferi* strains were cultured in Barbour-Stoenner-Kelly (BSK II) medium supplemented with 6% rabbit serum (PelFreez Biologicals, Rogers, AZ) and appropriate antibiotics (streptomycin, 50 μg/mL; kanamycin, 200 μg/mL; blasticidin, 10 μg/mL) and IPTG (0.01-10 mM) at 35 °C under 2.5% $CO_2$[95]. Infectious *B. burgdorferi* clone B31-A3-68-LS, which lacks linear plasmid lp56 and circular plasmid cp9, and has *flgBp-lacI* inserted in the restriction-modification gene *bbe02* gene on lp25[63,96], was the wild-type strain used in this study. Cloning vectors were propagated in *E. coli* strain TOP10 (Invitrogen, Carlsbad, CA). All *B. burgdorferi* strains and derivatives, and plasmids utilized in this study are described in Supplementary Data 1: Strains and plasmids.

### Assembly of constructs and transformation of *B. burgdorferi*
*B. burgdorferi* was transformed by electroporation as previously described[97]. Competent *B. burgdorferi* were freshly prepared from an exponential phase culture and electroporated with 15-30 μg of plasmid DNA. Transformants were confirmed through PCR and sequencing, and total plasmid content was determined[98].

All primers used in this study are listed in Supplementary Data 1: Primers. To generate the IPTG-inducible *bbd18* construct, the *B. burgdorferi flgB*$_p$-driven kanamycin resistance cassette[99] was cloned upstream of the *flacp* inducible promoter on the pTA*flacp* plasmid[63,64]. Fragments comprising 590-base pairs (bp) upstream of *bbd18* and the *bbd18* coding region were PCR-amplified and cloned flanking the *flgB*$_p$-driven kanamycin resistance cassette and *flacp* such that the IPTG-inducible promoter would drive transcription of *bbd18*, yielding the pTA*ibbd18* plasmid. The genes encoding ampicillin and kanamycin resistance on the vector backbone of pTA*ibbd18* were removed by digestion with *Msc*I and *Sca*I. The resulting *E. coli* construct was confirmed to be ampicillin-sensitive prior to *B. burgdorferi* transformation. B31-A3-68-LS (wt) was electroporated with pTA*ibbd18* and plated on solid media containing streptomycin, kanamycin, and 1 mM IPTG to generate the flacp::ibbd18 strain (Fig. 1A).

The cp32 plasmids were sequentially displaced with incompatible shuttle vectors carrying a cp32 fragment encompassing the *nucP* homolog through the PFam32/*parA* homolog (e.g. *bbp29-bbp32* of cp32-1; 3,792 bp); importantly, this fragment lacks the PFam49/*parB* plasmid maintenance gene (e.g. *bbp33* of cp32-1). The pBSV2G shuttle vector was digested with XmnI to remove the cp9 plasmid partition and replication region, generating the vector backbone pOG. Each amplified cp32 replication region was cloned into the pOG vector backbone to create 7 different cp32-displacing shuttle vector constructs. A single cp32-displacing shuttle vector was transformed into wt and flacp::ibbd18 cells to displace the targeted cp32 plasmid (eg. pOG::*bbp29-bbp32* would displace cp32-1). Gentamicin was utilized to select transformants, and cp32 displacement and plasmid maintenance were determined by multiplex PCR of selected clones as previously demonstrated[98,100]. The cp32-displacing shuttle vectors are unstable due to the lack of a PFam49/*parB* gene. Therefore, the cp32-displacing shuttle vector could be lost in clones lacking the targeted cp32 plasmid(s) through sequential passage in media lacking gentamicin. Once lost, a different cp32-displacing shuttle vector was transformed into spirochetes lacking one or more cp32 plasmid(s). This was repeated until all seven cp32 plasmids had sequentially been displaced from wt and flacp::ibbd18 cells (Fig. 7).

### Assessment of bacterial growth
IPTG-dependence of flacp::ibbd18 cells was assessed by growth in culture. Wild type and flacp::ibbd18 spirochetes were grown in medium supplemented with appropriate antibiotics and 1 mM IPTG, pelleted by centrifugation, washed twice in BSK-H media (Sigma-Aldrich, Atlanta, GA) and resuspended in fresh BSKII media with appropriate antibiotics at a density of $1 \times 10^5$ spirochetes/ml. IPTG was added to these cultures at final concentrations of 0, 0.01, 0.1, 1, or 10 mM. Cultures were grown in triplicate and counted every 24 h by dark-field microscopy using a Petroff-Hausser chamber, and by plating in media containing 1 mM IPTG if cell density was too low for reliable Petroff-Hausser counts. Additionally, live/dead cell staining with the LIVE/DEAD BacLight Bacterial Viability Kit (Invitrogen) was used per the manufacturer's instructions to visualize cells grown in the presence or absence of IPTG. Cells were imaged on a Nikon E80i fluorescent microscope under 40X magnification using a FITC channel to illuminate SYTO 9-labeled live cells and a TRITC channel to illuminate propidium iodine-labeled dead cells. Images were merged using FIJI[101], randomized, and counted.

### PacBio sequencing of *B. burgdorferi*
Genomic DNA purified from wt, flacp::ibbd18, and flacp::ibbd18(rec) cells grown in the presence and absence of IPTG was subjected to PacBio SMRT-sequencing. Briefly, gDNA was used to generate 16 barcoded SMRTbell libraries (Adapter kits 8 A & 8B) and subjected to SMRT-sequencing following manufacturer's instruction for Multiplex Microbial SMRTbell Libraries v2. (Pacific BioSciences, Menlo Park, CA)

with the following modifications. Two pools were independently generated for sequencing on two SMRT cells on the PacBio Sequel platform (Pacific BioSciences, Menlo Park, CA). Primer and polymerase were annealed to the first pool, which consisted of all 16 libraries following nonsize selection protocol. The second pool was the same except libraries were generated following the optional size selection protocol. Each SMRT cell underwent diffusion loading with a pre-extension time of 120 min and 10 hr movie time. The nonsize selected pool was loaded at 4.5pM, while the size-selected pool was loaded at 7pM. For each SMRT cell, SMRT Link RunQC showed a P1 value >50% with N50 longest subread of 9250 bp and 12750 bp, respectively. Reads were processed and mapped to the appropriate references using the pbsmrtpipe ds_modification_detection and sa3_ds_resequencing_fat pipelines (Pacific BioSciences).

### DNA and RNA isolation, sequencing, and quantitative real-time PCR

Total RNA was extracted from *B. burgdorferi* at mid-log phase ($5-7 \times 10^7$) or grown for 40 hrs in the absence of IPTG using TRIzol reagent (Life Technologies, Carlsbad, CA) per the manufacturer's instructions, and treated with 1 unit of DNAse I (Ambion, Foster City, CA) for 10 minutes at 37 °C. RNA was then quantified and subjected to Agilent Bioanalyzer 2200 Tape Station (Agilent, Santa Clara, CA) quality assessment. RNAs possessing RIN values ≥ 7.4 were used for downstream analysis.

For plasmid copy number qPCR, the number of spirochetes in culture was determined through counting by dark-field microscopy using a Petroff-Hausser chamber and cell lysates containing 5000 spirochetes/mL were used for qPCR analysis. Total genomic DNA was extracted from mouse tissues (heart or ear) utilizing the Zymo Research *Quick*-DNA/RNA miniprep plus kit (Irvine, CA) per the manufacturer's instructions. RNA was extracted from infected ticks by mechanical disruption and subsequent purification with TRIzol reagent or the Zymo Research *Quick*-DNA/RNA miniprep plus kit.

For RNA-seq, ribosomal RNA was depleted using the QIAseq FastSelect -5S/16 S/23 S kit (QIAGEN Sciences Inc., Germantown, MD) prior to first strand cDNA synthesis and NGS library preparation using the TruSeq Stranded mRNA-Seq Sample Preparation Kit (Illumina, Inc, San Diego, CA). The Illumina index adapters were diluted 2-fold for the ligation as recommended by the QIAseq protocol. Final library size distribution was assessed on a BioAnalyzer DNA 1000 chip (Agilent Technologies, Santa Clara, CA). The average size of the libraries was on target at around 350 bp. Libraries were quantified using the Kapa SYBR FAST Universal qPCR kit for Illumina sequencing (Kapa Biosystems, Boston, MA) on the CFX384 Real-Time PCR Detection System (Bio-Rad Laboratories, Inc, Hercules, CA). The libraries were diluted to 4 nM stocks and pooled equitably for sequencing. The 4 nM pool of libraries was prepared for sequencing by denaturing and diluting to a 1.5 pM stock for clustering to the flow cell. Libraries were sequenced on an Illumina NextSeq 550 at 2 ×75 bp paired-end using a Mid Output 150 cycle kit (Illumina, Inc, San Diego CA). The average cluster density was 177 k/mm2 resulting in ~140 M reads passing filter.

RNA-seq reads were compiled and filtered to remove any reads with PHRED scores less than 10 and aligned to the *B. burgdorferi* B31 genome (RefSeq AE000783-AE000794 and AE001575-AE001584) using bowtie2[102]. Reads for annotated genes were determined using featureCounts[103,104]. Differential expression analysis was assessed with edgeR[105] and DESeq2[106].

To generate cDNA, 1 µg of RNA was reverse-transcribed using the High Capacity cDNA Reverse Transcriptase kit (Life Technologies), per the manufacturer's instructions. qPCR and qRT-PCR reactions were performed using IQ™ SYBR® Green Supermix (Bio-Rad Life Sciences, Hercules, CA) with gene specific primer sets (500 nM) (Supplementary Data 1: Primers). Reactions were performed such that each experiment contained biological and technical triplicates on a Viia7 Real-Time PCR

System (Applied Biosystems, Foster City, CA) and analyzed with PRISM software (PRISM). Negative control reactions with primers lacking a template and RNA samples that underwent cDNA reactions in the absence of reverse transcriptase were included with each reaction to ensure Ct values were not obtained from primer-primer interactions or from contaminating genomic DNA. Additionally, the melt-curves were analyzed for each reaction. The specificities of primers for individual cp32 plasmids were also verified through qRT-PCR of flacp::ibbd18 cells lacking each cp32 plasmid (Supplemental Fig. S12).

### Sample preparation of *B. burgdorferi* Lysates for LCMS

*B. burgdorferi* were grown to mid-log phase ($5-7 \times 10^7$ spirochetes/ml), or for 40 hrs in the absence of IPTG, pelleted and rinsed 4 times with ice cold HN buffer. Pellets representing $5 \times 10^9$ spirochetes were then resuspended in 50 mM HEPES (pH 8.5) and 6 M guanidinium hydrochloric acid and vortexed until dissolved. The cells were then sonicated two times for 5 minutes each and stored at -80 °C until TMT mass-spec was performed. The mass spectrometry proteomics data have been deposited to the ProteomeXchange Consortium via the PRIDE[107] partner repository with the dataset identifier PXD037736.

Protein concentrations of diluted lysates were determined by BCA assay. Four hundred micrograms of protein in 6 M guanidinium HCl from each sample was reduced and alkylated using 5 mM DTT, followed by 15 mM iodoacetamide. Samples were diluted with 50 mM HEPES, pH 8.5 to a guanidinium concentration of 2.5 M. Two micrograms of LysC protease were added and the samples incubated at 37 °C for 8 hours. The samples were diluted with 100 mM HEPES, pH 8.0 to a concentration of 1.5 M guanidinium and incubated with 8ug of trypsin for 15 hours at 37 °C. The pH was adjusted to 2.0 with TFA and the samples were desalted and concentrated on 1cc Oasis HLB cartridges on a vacuum manifold. The eluted peptides were dried under vacuum and dissolved in 100ul of di water. The peptide concentrations were determined by fluorescent assay (Pierce Quantitative Peptide Assay) and 100ug of peptides from each sample were labeled with isobaric tags from a TMT-6plex reagent set (ThermoFisher) in 40ul reactions containing 100 mM TEABC, pH 8.5, and 40% acetonitrile. Labeling was determined to be ≥95% by mass spectrometry. The reactions were quenched with hydroxylamine, combined and the volume was reduced 50% under vacuum. Three hundred and fifty microliters of 0.1% TFA was added and the pH adjusted to 2.5 with the addition of 10% TFA. The mixture was desalted and concentrated as described. The peptides were dissolved in 50 mM TEABC, pH 8.5 and were loaded on to a 2.1 x 50 mm $C_{18}$ Extend HPLC column equilibrated in 5 mM TEABC-3% AcCN and developed with an acetonitrile gradient to 50%. Seventy fractions were collected and consolidated into five fraction pools, which were lyophilized. The residue was dissolved in 0.1% FA-3% AcCN, the peptides were quantitated (Pierce Colorimetric Assay) and adjusted to 100 ng/ul for injection.

For acquisition, the peptide samples were injected on to a Pepmap 100 C18, 2 µm, Φ 75 µm x 25 cm nano column (ThermoFisher) equilibrated in 0.1% formic acid developed with a gradient to 60% of 80% AcCN-20% 0.1% formic acid over 120 minutes at 300 nl/min. This was followed by a 5 minute wash at 80% AcCN. Acquisition was done on a Lumos Orbitrap mass spectrometer at 2000V, precursor selection 400-2000 m/z at 10ppm mass tolerance, 120,000 resolution. MS2 was CID, ion trap detection, in data dependent mode with an activation energy 35%, 0.7 m/z isolation and precursor selection from 400 to 1500 m/z. MS3 was HCD, OrbiTrap detection, at 65% collision energy with a scan range of 100 to 500 m/z, MS1 and MS2 isolation windows of 2 m/z, synchronous precursor selection and an Orbitrap resolution of 50,000.

All data were processed using Proteome Discoverer v2.4 (Thermo Scientific) with a SEQUEST HT search against the Uniprot KB *Borrelia burgdorferi* Proteome [taxon id 139] (03/2020) and common contaminants (theGPM.org), using a 10 ppm precursor mass tolerance and

a 0.6 Da fragment tolerance. Dynamic modifications included in the search were limited to oxidation [M], acetylation [Protein N-terminal], Met-loss [Protein N-terminal], and Met-loss + acetylation [Protein N-terminal], while carbamidomethylation [C] and TMT 6plex [K/ Peptide N-term] were the only static modifications utilized. Peptides and proteins were filtered at a 1% FDR using a target-decoy approach with a 2 peptide per protein minimum. Normalization was performed using total intensity of all peptides approach and corrections for isotope impurities in the TMT lots were performed [Lots TC256667, UK288831, VF291489].

## Phage quantification and purification

Wild-type and flacp::ibbd18 spirochetes were propagated for 24 hrs in fresh BSKII medium supplemented with appropriate antibiotics with and without 1 mM IPTG. To quantify DNA present after DNase treatment of culture supernatant, cultures were centrifuged and spirochetes were pelleted and resuspended in incomplete BSKII lacking BSA and rabbit serum supplemented with antibiotics, with and without 1 mM IPTG. Spirochetes were maintained in this deficient BSKII medium for 24 hrs prior to centrifugation to collect the culture supernatant. The supernatant was then centrifuged an additional two times and extracted with an equal volume of chloroform prior to DNaseI (Ambion) treatment at 37 °C for 1 hr. Supernatants were then re-extracted with an equal volume of chloroform prior to qPCR. These treated supernatants were passed into BSKII medium to ensure that no viable spirochetes remained. qPCR was performed on DNase-treated supernatant with chromosomal and plasmid primers. All qPCR reactions of *Borrelia* supernatants failed to detect the chromosomal *flaB* control, indicating that no unprotected cellular DNA remained after treatments.

For phage purification, wt and flacp::ibbd18 spirochetes were grown in BSKII medium supplemented with appropriate antibiotics and 1 mM IPTG to a density of ~5 × 10⁶ as measured with a Petroff-Hauser counting chamber. Spirochetes were pelleted, washed twice in BSK-H media (Sigma-Aldrich) and resuspended in fresh BSKII with appropriate antibiotics, with or without the addition of IPTG. Cultures were maintained for 48 hrs prior to centrifugation and PEG-precipitation or column purification of phage from cell-free culture supernatant as previously described[75]. Following PEG-precipitation, samples were treated with DNaseI (Ambion) at 37 °C for 1 hr. Samples were then either concentrated further with additional PEG-precipitations or DNA was extracted following previously described protocols[75]. Phage DNA was dyed with GelRed® (Biotium, Fremont, CA) and visualized on a 0.4 agarose gel run at 20 V overnight and then at 40 V for several hours.

Phage purification for EM imaging involved PEG precipitation and chloroform extraction of centrifuged supernatant. This was followed by purification with 100 KDa MWCO centrifugal filters.

## Transmission electron microscopy

5 μl droplets of *B. burgdorferi* cells or phage were absorbed to the surface of freshly glow-discharged, formvar-coated 200 mesh copper grids and then negatively stained with 5 μl droplets of 2% methylamine vanadate (Nanoprobes, Yaphank, NY) prior to viewing on a Hitachi H-7800 transmission electron microscope at 120 kV (Hitachi-High-Technologies Corporation, Tokyo, Japan).

## Experimental mouse-tick infection studies

Mouse infections were conducted with 6- to 8-week female RML mice, a derivative of Swiss-Webster mice reared at the Rocky Mountain Laboratories breeding facility. In total, 16 mice were used for needle inoculations and/or larval feedings and 18 mice were used for nymph feedings for a total of 34 mice. Mice were housed at ambient humidity of 50%±10%, at ambient temperature between 20.6 and 23.9 °C, and under a 12 hr ON/12 hr OFF light cycle. The omission of male mice has no effect on this study as both sexes are equally susceptible to *B. burgdorferi* infections. All mice were monitored daily by trained veterinary staff, and any abnormalities were reported and further monitored. No mice in this study had to be euthanized prior to the set endpoint. Mice were co-housed unless undergoing a larval or nymph feeding, during which time they were housed singly to prevent tick removal by cage mates. Once all ticks had fed to repletion, mice were again co-housed. At the set endpoint, all mice were euthanized by cervical dislocation under isoflurane anesthesia. All animal work received approval from the Rocky Mountain Laboratories, NIAID, NIH Animal Care and Use Committee.

Prior to injection of mice, the complete plasmid profiles of *B. burgdorferi* inocula were determined to ensure they retained plasmids lp25, lp28-1, and lp36, which are required for infectivity but unstable[100,108,109]. Mice were inoculated intraperitoneally (4 × 10³ spirochetes) and subcutaneously (1 × 10³ spirochetes), with the number of injected spirochetes pre-determined by Petroff-Hausser counting and confirmed by plating aliquots of the inocula. Infection of mice by *B. burgdorferi* was assessed at 3 weeks post-injection by immunoblot analysis for seroconversion to *B. burgdorferi* antigens and at 5 weeks post-injection through isolation of spirochetes from mouse tissues (ear, bladder, and fat) in BSKII medium containing appropriate antibiotics and 1 mM IPTG. These experiments were repeated two times. The first experiment involved needle inoculation of 4 mice with wt and 4 mice with flacp::ibbd18 spirochetes. Following positive serology at three weeks, mice were then fed upon by larvae. Larvae were allowed to molt to nymphs and 5 naïve mice were fed upon by ~20 wt-infected nymphs/mouse and 5 naïve mice were fed upon by ~20 flacp::ibbd18-infected nymphs/mouse. All mice, including those fed upon by nymphs, were positive by tissue isolation at 5 weeks post-exposure. The second experiment involved needle inoculation of 3 mice with wt and 3 mice with flacp::ibbd18 spirochetes. All mice sero-converted at 3 weeks and were then fed upon by larvae. Larvae were allowed to molt to nymphs and 3 naïve mice were fed upon by ~20 wt-infected nymphs/mouse and 3 naïve mice were fed upon by ~20 flacp::ibbd18-infected nymphs/mouse. All mice, including those fed upon by nymphs, were positive by tissue isolation at 5 weeks post-exposure.

Larval *Ixodes scapularis* were purchased from Oklahoma State University. *I. scapularis* were maintained between feeds in a temperature- and humidity-controlled chamber (Caron Model 7000-25) or in bell jars containing potassium sulfate-saturated water at room temperature. Approximately 100-200 naïve *I. scapularis* larvae were fed to repletion per infected mouse. Acquisition and retention of *B. burgdorferi* by larval ticks was assessed 1 week, 7 weeks and ~16 weeks (once fully molted) after drop-off, and spirochete load was determined through mechanical disruption and plating. Naïve mice were fed upon by 15-20 infected *I. scapularis* nymphs. The number of viable *B. burgdorferi* in nymphs was assessed prior to feeding, during feeding, at repletion and 24 hrs, 48 hrs, 10 days, and 1 month after drop-off through mechanical disruption and plating.

*I. scapularis* larvae were infected with ΔrpoS spirochetes by immersion[110] because spirochetes lacking *rpoS* are not infectious for mice and therefore cannot be acquired by feeding[17]. Briefly, larvae were dehydrated with ammonium sulfate for 48 hrs prior to placing in 1 ml of either ΔrpoS or flacp::ibbd18ΔrpoS cultures with ~1 × 10⁸ spirochetes per ml. The larvae were incubated with the spirochetes for one hour at 35 °C before centrifuging at 1,500 x g for 1 min and washing two times in PBS. Artificially infected larvae were then placed upon naïve mice (one mouse per strain) and allowed to feed to repletion. Following the molt of these replete larvae to nymphs, the nymphs were placed upon naïve mice (one mouse per strain) and fed to repletion. The number of viable *B. burgdorferi* in nymphs was assessed prior to feeding, at repletion and 24 hrs, 48 hrs, 10 days, and 1 month after drop-off through mechanical disruption and plating.

## Immunofluorescence assays of tick midguts

Immunofluorescence assays (IFAs) of tick midguts were performed as previously described[61]. Briefly, unfed or replete *I. scapularis* nymphs were disrupted in PBS on glass microscope slides. The tick midgut was transferred to a new slide, uniformly disrupted and allowed to air dry. These slides were then heat-fixed and permeabilized in an equal volume of methanol and acetone for 30 minutes. Mouse monoclonal antibody H5332 and rabbit polyclonal antisera were used to detect all OspA- and OspC-expressing spirochetes, respectively[111]. Primary antibodies were detected using rhodamine-conjugated anti-mouse (OspA) and FITC-conjugated anti-rabbit (OspC) secondary antibodies. All antibodies were diluted 1:100 in PBS with 0.75% Bovine Serum Albumin (BSA) and incubated at 25 °C for 30 minutes. Vectashield mounting medium (Vector Laboratories, Burlingame, CA) was used to prevent the photobleaching of fluorochromes. Images were captured on a Nikon E80i fluorescent microscope under 40X magnification using a FITC channel to illuminate OspC-expressing cells and a TRITC channel to illuminate OspA-expressing cells. Images were merged using FIJI[101].

## Reporting summary

Further information on research design is available in the Nature Portfolio Reporting Summary linked to this article.

## Data availability

All RNA-seq data generated in this study are available at NCBI GEO at accessions GSM6278774- GSE207123. Source data are provided with this paper. The TMT mass-spec proteomics data generated in this study have been deposited to the ProteomeXchange Consortium via the PRIDE[107] partner repository with the dataset identifier PXD037736. Source data for all graphs are provided with this paper.

## Code availability

PacBio reads were processed and mapped to the *Borrelia burgdorferi* B31 genome (RefSeq AE000783-AE000794 and AE001575-AE001584) references using the pbsmrtpipe ds_modification_detection and sa3_ds_resequencing_fat pipelines (Pacific BioSciences). RNA-seq reads were compiled, parsed, and filtered to remove any reads with PHRED scores less than 10 and aligned to the *Borrelia burgdorferi* B31 genome (RefSeq AE000783-AE000794 and AE001575-AE001584) using bowtie2 v2.2.5. Alignments were then sorted with samtools v1.2 prior to utilizing featureCounts v1.5.0-p3 to determine reads for annotated genes and small RNAs identified by Popitsch, et al.[73]. Differential expression was determined with DeSeq v1.14.1 and edgeR v2.6.0. Tandem mass-tag mass-spectrometry was performed on a Lumos Orbitrap mass spectrometer and all data was processed with Proteome Discoverer v2.4 with a SEQUEST HT search against the Uniprot KB *Borrelia burgdorferi* Proteome [taxon id 139]. Fluorescent microscopy images were taken with a Nikon E80i fluorescent microscope and image files were obtained using Nikon Elements version 4.2. Transmission electron microscopy images were taken with a Hitachi H-7800 transmission electron microscope. Images were prepared with FIJI version 2.1.0.

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

## Acknowledgements

We would like to acknowledge Stacy Ricklefs and Dan Bruno for their help with RNA sequencing and Elizabeth Fischer for help with electron microscopy. We would also like to thank Ashley Groshong and Liam Fitzsimmons for their careful reviews and helpful suggestions for this manuscript. This research was supported by the Intramural Research Program of the National Institute of Allergy and Infectious Diseases, National Institutes of Health. PRS is supported by NIH grants R21AI151597 and P30GM140963. MK and PRS are supported by Montana INBRE (P20GM103474).

## Author contributions

J.W., B.C, C.H., V.C., D.D., K.B., G.N., M.K and P.R. performed the experimental studies. J.W., C.M., L.R.O., P.S. and P.R. analyzed the data. J.W. and P.R. prepared and edited the manuscript. J.W., P.S. and P.R supervised the work.

## Competing interests

The authors declare no competing interests.
