## [Peer Review File · Nature Communications]

Coupled induction of prophage and virulence factors during tick transmission of the Lyme disease spirocheteREVIEWER COMMENTS

Reviewer #1 (Remarks to the Author):

In *Borrelia burgdorferi*, the alternative sigma factor RpoS is critical for bacterial adaptation during host infection. Prior studies have determined that BBD18 is a negative regulator of RpoS, but they were unable to inactivate *bbd18*. In the current study the authors generated an inducible *bbd18* gene at the endogenous plasmid locus and demonstrated the essential nature of BBD18 for viability of wild-type spirochetes in vitro. Their finding that *bbd18* was not required for murine infection agreed with previous studies, but they expanded this work to now investigate the role of *bbd18* in vector colonization and transmission. Larval ticks acquired the *ibbd18* spirochetes from infected mice, retained them through the nymphal molt, and transmitted the mutant spirochetes to naïve mice. The number of viable *ibbd18* spirochetes in nymphs then dropped sharply after feeding and continued to decrease until viable spirochetes could no longer be detected. Numerous RpoS and RpoS-dependent genes were induced following BBD18 depletion and induction resulted in spirochete lysis. The authors also found that RpoS regulates phage lysis-lysogeny with differential expression of plasmid prophage genes and phage particles being detected in the bacterial supernatants after induction. This is the first report of a mechanistic link between endogenous transducing prophages and the RpoS-dependent adaptive response of the Lyme disease spirochete. Overall, this is a very well-written manuscript and experimental approach is technically sound. The authors do an excellent job of presenting the significance and relevance of their findings. I have no concerns with the work and recommend an "Accept" decision.

Reviewer #2 (Remarks to the Author):

This far-ranging study by Wachter et al. utilizes a diversity of sophisticated genetic, transcriptomic, tick-mouse modeling, and biochemical approaches to investigate BBD18, a protein essential for in vitro growth, that negatively regulates RpoS, itself the major regulator for the transition of Bb from the feeding tick to the vertebrate host. They make several exciting findings, including (a) BBD18 essential function is to modulate *rpoS* expression; (b) nearly all of the effects of BBD18-deficiency can be explained by its effect on *rpoS*; (c) in the absence of *bbd18*, overexpression of *rpoS* results in induction of multiple members of the cp32 family of prophages, as well as a (likely) lp28-2 prophage, thus explaining the essential nature of *bbd18* for in vitro survival; and (d) *bbd18* is not essential for productive transmission of the spirochete to the vertebrate host, but is essential for survival in the tick post-feeding.

Given that the authors' findings touch on several interesting biological questions regarding the Lyme spirochete, they should pay meticulous attention to present their data in a systematic fashion, clearly articulating the specific biological question each data set addresses. This would be fostered by explicitly raising some of the above questions in the Introduction or as an introductory sentence (or sentences) at the start of subsections in Results. Examples include:

1. Lines 375-383. The first reference to Fig. S11 occurs in Discussion. This information gives important context to the reader because the question of how BBD18 functions was raised several times earlier in the manuscript. These observations should be presented in Intro or Results.
2. Clearly a central question addressed by the authors is the degree to which the effects of BBD18-deficiency are solely due to the unregulated expression of *rpoS*. There are parts of the manuscript in which the data relevant to this question can be more clearly presented. Examples include:
 - a. Fig. S1B indicates that only a small number of (e.g., 15-17) genes are positively regulated by BBD18 in the absence of RpoS. Is anything known about these genes, e.g., are any known regulators, or known to have expression patterns similar to *rpoS*? I think this is partially addressed in Lines 198-201, but a systematic presentation of these complex data followed by a step-by-step discussion of their implications would be helpful. I found myself wondering about the similarity or differences in RNA-seq gene set data for wild type, induced *flacp::ibbd18*, uninduced *flacp::ibbd18*, and *flacp::ibbd18 ΔrpoS*.
 - b. Line 357-358. What do the authors make of the observation that BBD18-regulated genes that are not regulated by RpoS are membrane proteins? Even if they have no ideas, stating so would be useful so that readers don't feel like they are missing something.
3. Line 397-399. The interesting concept that some genes are specifically required for survival after the nymphal blood meal would be better highlighted if it were articulated in the Intro or the

appropriate section of Results.

4. Lines 359-374. The issue of *bbd21* and the conundrum of *bbd18* loss in the context of a larger deletion is more clearly laid out here than in Results.

5. Line 164-172. Inclusion of background about *ittA* (refs. 67-68) at the beginning of this section would be helpful, as would some interpretation/speculation concerning the absence of *ittA* RNA.

6. Line 416. Highlighting the potential of *lp28-2* as encoding a phage unrelated to *cp32* phage in the Results section would be helpful in a reader's interpretation of the data. (Future experiment, not required for this study, would be to use immunoEM to determine if *lp28-2* encodes one of the two morphological types of phages observed.)

The authors should address the following points:

6. Lines 328-331 and Fig. 8B, D and legend. (a) Fig. 8B labeling of fraction of nymphs infected should be made more explicit in panel and in legend; (b) a prediction is that deletion of *rpoS* may rescue the defect in survival of *flacp::ibbd18* in fed nymphs. By simply determining fraction of infected nymphs (comparing Fig. 8B, right and 8D, right), it is not clear that this is the case. Perhaps CFU or qPCR determination of numbers of spirochetes would be necessary to demonstrate this point.

7. Line 233. Can you either sequence or PCR to test if the DNA is in fact *cp32* and if so, which *cp32*? A possible outcome is that *cp32-7* and *cp32-9* are not represented in this DNA.

8. Figures 3A and 4F. Does *RpoS* have inhibitory effects on *bbd18* expression? If the data in Figure 3A and 4F can be directly compared, the *bbd18* expression in WT is ~7 *bbd18* transcripts / 1000 *flaB* transcripts (Figure 3A), while the *bbd18* expression in $\Delta rpoS$ is ~25 *bbd18* transcripts / 1000 *flaB* transcripts (Figure 4F).

9. Figure 5A. The copy number for the *cp32* derivatives are consistently lower than 1 copy per chromosome even in WT, which appears to be even lower than the other plasmids (lower panel) – is there any explanation for this?

10. The difference in *bbd18* transcripts in *flacp::ibbd18* cells in the presence or absence of IPTG, although intuitively clear, did not reach statistical difference. This should be noted (e.g., consider including some caveat(s) to definitive statements such as “Unmodulated expression of *rpoS* (through *bbd18* repression) is responsible for *B. burgdorferi* lysis in vitro”).

Minor points

11. Line 171. Slightly elevated compared to what? uninduced *flacp::ibbd18* or wt?

12. Line 218. Specify which *cp32* increased and which did not.

13. Line 277. Given that *rpoS* is down-regulated in the mammalian host (Livengood, 2008), the authors should consider tempering the inclusion of “Surprisingly, ...”.

14. Line 278, Fig. 8 legend. Provide the time post-infection that the mice were evaluated for infection.

15. Lines 285-286. The authors should note that the higher bacterial burden in *flacp::ibbd18*-infected relative to wt-infected mice was restricted to the ear and not found in the heart (Figure 8A).

16. Line 389. “*rpoS* expression or *RpoS* action”

17. Fig. S6 legend title: “qRT-PCR of *ittA* (not “total”) RNA”

18. Fig. 5C legend. Which *cp-32* transcripts were measured?

19. Line 106. Add “indistinguishable in growth rate from wt”

20. Lines 146-47: “Most” (not “Many”)... “were” (not “are”).

21. Lines 188-190. Add “(Supplemental Table)” to the end of the following sentence “Although de-induction of *flacp::ibbd18* led to increased expression of around 300 genes, there were no genes with increased expression following de-induction of *flacp::ibbd18* $\Delta rpoS$ when compared to induced *flacp::ibbd18* $\Delta rpoS$.”

22. Lines 208-209. “Collectively, these data indicate that the level of BBD18 affects expression of genes on the right telomeric end of *lp17*, and that BBD18 either directly, or indirectly through an unknown mechanism, transcriptionally regulates *rpoS*”.

**RESPONSE to REVIEWER COMMENTS**

Reviewer #1 (Remarks to the Author):

**We are pleased that Reviewer #1 found our study complete, well-written and acceptable for**
**publication in *Nature Communications* without further revision.**

In *Borrelia burgdorferi*, the alternative sigma factor RpoS is critical for bacterial adaptation
during host infection. Prior studies have determined that BBD18 is a negative regulator of RpoS,
but they were unable to inactivate *bbd18*. In the current study the authors generated an
inducible *bbd18* gene at the endogenous plasmid locus and demonstrated the essential nature
of BBD18 for viability of wild-type spirochetes in vitro. Their finding that *bbd18* was not
required for murine infection agreed with previous studies, but they expanded this work to now
investigate the role of *bbd18* in vector colonization and transmission. Larval ticks acquired the
*ibbd18* spirochetes from infected mice, retained them through the nymphal molt, and
transmitted the mutant spirochetes to naïve mice. The number of viable *ibbd18* spirochetes in
nymphs then dropped sharply after feeding and continued to decrease until viable spirochetes
could no longer be detected. Numerous RpoS and RpoS-dependent genes were induced
following BBD18 depletion and induction resulted in spirochete lysis. The authors also found
that RpoS regulates phage lysis-lysogeny with differential expression of plasmid prophage
genes and phage particles being detected in the bacterial supernatants after induction. This is
the first report of a mechanistic link between endogenous transducing prophages and the
RpoS-dependent adaptive response of the Lyme disease spirochete. Overall, this is a very well-
written manuscript and experimental approach is technically sound. The authors do an
excellent job of presenting the significance and relevance of their findings. I have no concerns
with the work and recommend an "Accept" decision.

Reviewer #2 (Remarks to the Author):

**We thank Reviewer #2 for their positive review of our manuscript and for their helpful and**
**constructive comments, which we address in full below.**

This far-ranging study by Wachter et al. utilizes a diversity of sophisticated genetic,
transcriptomic, tick-mouse modeling, and biochemical approaches to investigate BBD18, a
protein essential for in vitro growth, that negatively regulates RpoS, itself the major
regulator for the transition of *Bb* from the feeding tick to the vertebrate host. They make
several exciting findings, including (a) BBD18 essential function is to modulate *rpoS*
expression; (b) nearly all of the effects of BBD18-deficiency can be explained by its effect on
*rpoS*; (c) in the absence of *bbd18*, overexpression of *rpoS* results in induction of multiple

members of the cp32 family of prophages, as well as a (likely) lp28-2 prophage, thus
explaining the essential nature of bbd18 for in vitro survival; and (d) bbd18 is not essential
for productive transmission of the spirochete to the vertebrate host, but is essential for
survival in the tick post-feeding.

Given that the authors' findings touch on several interesting biological questions regarding
the Lyme spirochete, they should pay meticulous attention to present their data in a
systematic fashion, clearly articulating the specific biological question each data set
addresses. This would be fostered by explicitly raising some of the above questions in the
Introduction or as an introductory sentence (or sentences) at the start of subsections in
Results. Examples include:

1. Lines 375-383 (now lines 456-464). The first reference to Fig. S11 occurs in Discussion.
This information gives important context to the reader because the question of how BBD18
functions was raised several times earlier in the manuscript. These observations should be
presented in Intro or Results.

Thank you for this helpful suggestion. We have added material throughout the manuscript
to introduce the motivation for each experiment and to summarize what we learned in
anticipation of the next section. This is most easily referenced with the highlighted
comparison of original and revised manuscripts.

With respect to this first example, we agree that the proposed mechanism by which BBD18
regulates *rpoS*/RpoS should have been addressed earlier in the manuscript. We therefore
added lines 153-159 in the Results section to introduce this point sooner, whereby
Supplemental Figure S11 became Supplemental Figure S4.

2. Clearly a central question addressed by the authors is the degree to which the effects of
BBd18-deficiency are solely due to the unregulated expression of *rpoS*. There are parts of
the manuscript in which the data relevant to this question can be more clearly presented.

Examples include:

a. Fig. S1B indicates that only a small number of (e.g., 15-17) genes are positively regulated
by BBD18 in the absence of RpoS. Is anything known about these genes, e.g., are any known
regulators, or known to have expression patterns similar to *rpoS*? I think this is partially
addressed in Lines 198-201, but a systematic presentation of these complex data followed
by a step-by-step discussion of their implications would be helpful. I found myself
wondering about the similarity or differences in RNA-seq gene set data for wild type,
induced *flacp::ibbd18*, uninduced *flacp::ibbd18*, and *flacp::ibbd18 ΔrpoS*.

We agree that a clearer analysis of the RNA-seq data is required to answer this central
question and have substantially revised the manuscript to address this concern. We have
added an additional section about the *ΔrpoS* transcriptome on Lines 223-270 and have
included further information about these 15 genes and their expression patterns relative to
RpoS. Additionally, through a bioinformatics approach we identified a conserved motif
directly upstream of 12/15 genes potentially directly regulated by BBD18. However, as this is
strictly an *in silico* prediction without any biochemical or genetic data, we have not
described this finding in the manuscript as further investigations are needed that are
outside of the scope of this study.

We have attempted to add a clearer interpretation of the implications of comparative RNA-
seq datasets of Supplemental Figure S1 with respect to regulation of RpoS versus RpoS-
dependent gene expression on lines 218-222. We also expanded our mention of lp17-
encoded genes in the presence/absence of *rpoS* and included our thoughts on the
mechanism of BBD18 regulation of *rpoS* on lines 338-350. However, we acknowledge that it
is challenging to describe comparisons between samples +/- BBD18 and +/- RpoS without
getting mired in the underlying experimental details. We therefore frequently refer to RNA-
seq data in pertinent sections of the Supplemental Table that better illustrate the points we
are making. All of these are highlighted in the text of the revised manuscript for comparison
with the original submission

b. Line 357-358. What do the authors make of the observation that BBD18-regulated genes
that are not regulated by RpoS are membrane proteins? Even if they have no ideas, stating
so would be useful so that readers don't feel like they are missing something.

Thank you for raising this point. What we intended to convey and now state is that
membrane-bound proteins are not likely candidates for cytoplasmic transcriptional
regulatory intermediates between BBD18 and RpoS. This is described on lines 245-248 of
the new section in the **Results** titled **Scrutinizing *ΔrpoS* transcriptomes for regulatory**
**intermediates** (lines 223-270) and lines 433-438 of the **Discussion**.

3. Line 397-399 (Now lines 478-480). The interesting concept that some genes are
specifically required for survival after the nymphal blood meal would be better highlighted if
it were articulated in the Intro or the appropriate section of Results.

We have articulated that down-regulation of *rpoS*/RpoS by BBD18 is critical for survival of *B.*
*burgdorferi* in the fed tick midgut on lines 388-390 of the **Results** where we state "... with no

viable spirochetes detected by one month after the nymphal blood meal (Figure 8b),
demonstrating that BBD18 is required for spirochete persistence in the tick midgut after the
nymphal bloodmeal" and reiterated on lines 482-487 of the **Discussion**.

4. Lines 359-374 (now lines 441-455). The issue of bbd21 and the conundrum of bbd18 loss
in the context of a larger deletion is more clearly laid out here than in Results.

Thank you for encouraging us to address this conundrum more fully. We have added
material to several places in the **Results** section (lines 140-141, 175-179, 249-270) to
describe the hypothetical or annotated functions of these lp17 genes and to place our data
in the context of published observations. We also added a new section to the **Results** (lines
338-350) entitled **Potential role of BBD21/ParA in prophage induction**.

5. Line 164-172 (now Lines 189-199). Inclusion of background about *ittA* (refs. 67-68) at the
beginning of this section would be helpful, as would some interpretation/speculation
concerning the absence of *ittA* RNA.

Thank you. We moved the line referencing the importance of *ittA* to the beginning of the
paragraph and added a clearer introduction to *Borrelia* sRNAs (line 183-187). While we do
not know what accounts for the disparity in *ittA* expression, between studies, it may reflect a
difference in the genome content of our wt strain with that used by Medina-Pérez, *et al.* This
is now stated on lines 197-199.

6. Line 416 (now Lines 496-497). Highlighting the potential of lp28-2 as encoding a phage
unrelated to cp32 phage in the Results section would be helpful in a reader's interpretation
of the data. (Future experiment, not required for this study, would be to use immunoEM to
determine if lp28-2 encodes one of the two morphological types of phages observed.)

Thank you. We have added material at pertinent points in the **Results** (lines 297-298, 299-
302, 324-337) to highlight the potential role of lp28-2 as a novel prophage morphologically
and genetically distinct from the defined cp32 prophage. We have replaced the gel in
Figure 6 because we identified an issue with the labeling of lanes on the original gel. The
band of phage DNA purified from the supernatant of lysed cells visualized on this gel
migrates between the 14kb and 50 kb markers and is therefore compatible with the size of
either prophage genome. We would like to identify the observed B31 phage morphotypes
in future work and hope to use immunoEM to do so.

The authors should address the following points:

6. Lines 328-331 (now Lines 406-408) and Fig. 8B, D and legend. (a) Fig. 8B labeling of
fraction of nymphs infected should be made more explicit in panel and in legend; (b) a
prediction is that deletion of *rpoS* may rescue the defect in survival of *flacp::ibbd18* in fed
nymphs. By simply determining fraction of infected nymphs (comparing Fig. 8B, right and
8D, right), it is not clear that this is the case. Perhaps CFU or qPCR determination of
numbers of spirochetes would be necessary to demonstrate this point.

We have altered the labeling in Fig 8B with bold type to more clearly demonstrate the
fraction of uninfected nymphs. However, a direct quantitative comparison of spirochete
burden as CFU or qPCR is not meaningful in this experiment because the nymphs analyzed
in Fig 8B were naturally infected by feeding on an infected mouse, while the nymphs
analyzed in Fig 8D were artificially infected with delta *rpoS* spirochetes by immersion. Still,
the qualitative difference in outcome, in which *flacp::ibbd18* spirochetes did not survive in
**any** of the unfed adult ticks analyzed, whereas *flacp::ibbd18ΔrpoS* spirochetes persisted in
**all** of them, supports our conclusion that unmodulated RpoS induction is responsible for the
loss of viable *flacp::ibbd18* spirochetes in infected ticks following the nymphal bloodmeal.

7. Line 233 (now Line 494). Can you either sequence or PCR to test if the DNA is in fact cp32
and if so, which cp32? A possible outcome is that cp32-7 and cp32-9 are not represented in
this DNA.

Theoretically yes, but co-authors Pat Secor and Margie Kinnersley are in the process of
sequencing the DNA to determine what is packaged in purified *B. burgdorferi* phage heads
and we do not want to compromise their study. We did perform qPCR of chloroform- and
DNase-treated supernatant to identify the presence of cp32 DNA in de-induced
*flacp::ibbd18* cultures (and Ip28-2 DNA), but as we did not sequence the DNA in Fig. 6A, we
cannot definitively state whether the DNA on the gel is indeed cp32 (or Ip28-2). Since the
primers used for qPCR of supernatant were intentionally designed to amplify all cp32s, we
can say that cp32 DNA is present in the DNase-treated supernatant of de-induced
*flacp::ibbd18* cultures, but we do not know if any single cp32 is more prevalent than others.
Previous analyses of cp32 phage by Christian Eggers demonstrated that in strain CA-11.2A,
cp32-3 is the plasmid that is most commonly packaged into phage heads (Eggers CH,
Kimmel BJ, Bono JL et al. Transduction by φBB-1, a bacteriophage of *Borrelia burgdorferi*. J

Bacteriol 2001;183:4771–8), which is consistent with the copy number of cp32-3 plasmids
detected in Figure 5. We mention that the size of the DNase-shielded DNA is consistent
with lp28-2 or cp32 phage genomes on line 293-294. Additionally, we state on lines 298-300
that one of the two phage types present in the supernatant of lysed flacp::ibbd18 cells is
similar in morphology to the cp32 ΦBB-1 phage.

8. Figures 3A and 4F. Does RpoS have inhibitory effects on bbd18 expression? If the data in
Figure 3A and 4F can be directly compared, the bbd18 expression in WT is ~7 bbd18
transcripts / 1000 flaB transcripts (Figure 3A), while the bbd18 expression in ΔrpoS is ~25
bbd18 transcripts / 1000 flaB transcripts (Figure 4F).

That is a good point; the *bbd18* transcript level in wt does differ from ΔrpoS in these qRT-
PCR data. However, we found comparable levels of *bbd18* transcript in wt and ΔrpoS
spirochetes by RNA-seq analysis. Therefore, we cannot conclude that RpoS has a
reproducible and significant inhibitory effect on *bbd18* expression.

9. Figure 5A. The copy number for the cp32 derivatives are consistently lower than 1 copy
188 per chromosome even in WT, which appears to be even lower than the other plasmids
(lower panel) – is there any explanation for this?

We were puzzled by this as well and repeated the qPCR analysis with purified DNA and
whole cell lysates. We found that freshly prepared whole cell lysates were more reliable than
purified DNA for plasmid quantitation by qPCR, but that many plasmids were consistently
detected at less than one copy per chromosome. We believe that this reflects the polyploidy
of *B. burgdorferi* during exponential growth in culture, with a chromosomal copy number
greater than one (Biorxiv – Takacs, N *et al.* Polyploidy, regular patterning of genome copies,
and unusual control of DNA partitioning in the Lyme disease spirochete). Hence a slightly
lower copy number of the cp32 plasmids relative to the chromosome could be explained by
this recent finding.

10. The difference in bbd18 transcripts in flacp::ibbd18 cells in the presence or absence of
IPTG, although intuitively clear, did not reach statistical difference. This should be noted
(e.g., consider including some caveat(s) to definitive statements such as “Unmodulated
expression of rpoS (through bbd18 repression) is responsible for *B. burgdorferi* lysis in
vitro”.

The inability to restore *bbd18* expression to wt levels, even at 10 mM concentrations of IPTG,
was noted on Lines 95-96. Hence the expression of de-induced *flacp::ibbd18 bbd18*
transcript is significantly lower than wt cells, but not statistically significantly different from
induced *flacp::ibbd18* cells. However, the biological significance of uninduced *bbd18*
expression is cell death, whereas the growth phenotype, transcriptome and proteome of
induced *flacp::ibbd18* cells are similar to wt cells, with only 25 genes up-regulated in
induced *flacp::ibbd18*, presumably due to the slightly elevated levels of *rpoS*, which we
conclude on Lines 141-143.

Minor points

11. Line 171 (now Line 194-197). Slightly elevated compared to what? uninduced
*flacp::ibbd18* or wt?

Thank you for pointing this out. We have added the line "compared to wt and induced
*flacp::ibbd18* cells" on Lines 195-196.

12. Line 218 (now Lines 277-280). Specify which cp32 increased and which did not.

We have now specified which cp32 plasmids were increased on Line 279.

13. Line 277 (now Line 354). Given that *rpoS* is down-regulated in the mammalian host
(Livengood, 2008), the authors should consider tempering the inclusion of "Surprisingly, ...".

We acknowledge and reiterate on Lines 530-533 that *rpoS* is down-regulated by another
means in the mammalian host. "Surprisingly" referred to our stated hypothesis that BBD18
would be required to modulate *rpoS* expression *in vivo* as well as *in vitro*; we have revised
the wording (Line 354) for clarity.

14. Line 278 (now Line 355), Fig. 8 legend. Provide the time post-infection that the mice
were evaluated for infection.

We have added the time post-feeding that the serological response and tissues were
analyzed on line 356 and to the Figure 8 legend.

15. Lines 285-286 (now Line 363). The authors should note that the higher bacterial burden

in flacp::ibbd18-infected relative to wt-infected mice was restricted to the ear and not found
in the heart (Figure 8A).

We have now indicated on line 364 and in the legend to Figure 8 that the higher bacterial
burden in flacp::ibbd18-infected mice relative to wt-infected mice was in the ear.

16. Line 389 (now Line 470). "rpoS expression or RpoS action"

This has been added to line 470.

17. Fig. S6 (now Fig S7) legend title: "qRT-PCR of ittA (not "total") RNA"

Thank you for catching this; we altered the wording in the Fig. S7 legend title.

18. Fig. 5C legend. Which cp-32 transcripts were measured?

We have now identified (Lines 859-860) which cp32 transcripts were increased in de-
induced flacp::ibbd18.

19. Line 106. Add "indistinguishable in growth rate from wt"

We have added this to Lines 106-107.

20. Lines 146-47 (now line 167): "Most" (not "Many")... "were" (not "are").

We have revised the wording on line 167.

21. Lines 188-190 (now Lines 214-217). Add "(Supplemental Table)" to the end of the
following sentence "Although de-induction of flacp::ibbd18 led to increased expression of
around 300 genes, there were no genes with increased expression following de-induction of
flacp::ibbd18ΔrpoS when compared to induced flacp::ibbd18ΔrpoS."

We have added "(Supplemental Table: drpoS up-regulated)" and referenced Figure S1.b on
lines 214-217.

22. Lines 208-209. "Collectively, these data indicate that the level of BBD18 affects

expression of genes on the right telomeric end of lp17, and that BBD18 either directly, or
indirectly through an unknown mechanism, transcriptionally regulates rpoS”.

This entire section was changed to elaborate on the lp17-encoded genes. This sentence is
no longer included at this point in the manuscript as a new section was added (Lines 223-
270) to introduce the importance of these genes earlier in the **Results** section.

Reviewer #3 (Remarks to the Author)

We thank Reviewer #3 for their positive review of our manuscript. We have amended the
manuscript in response to the thoughtful comments raised by Reviewer #3 as highlighted in
the revised manuscript and addressed below.

This report from Wachter et al. describes the linkage of BBD18 production to RpoS synthesis
and how this affects borrelial viability and, interestingly, the production of bacteriophage.
The study is well-executed and controlled for by several inducible strains in a broad range of
global readouts. The sequential elimination of the cp32s is impressive given the iterative
process that it involved, and the inherent difficulties associated with borrelial genetics.
Importantly, the resulting cp32-deficient strain is tested for its survival and phage
production phenotypes when BBD18 levels are reduced and RpoS levels concomitantly
increase. The requirement for BBD18 was retained unless rpoS was inactivated directly
linking RpoS to the phenotype observed. While the RpoS-associated prophage induction is
clearly outlined, one issue that is problematic is the repeated description of the phage
observed being a transducing phage. The only data shown indicates that it packages cp32
DNA.

While other groups have demonstrated this effect in *B. burgdorferi*, the phage evaluated
here was not tested in this manner. To assume that the transducing activity is associated
with the observed phage is likely but not formally proven.

This is an accurate assessment, as we did not demonstrate the transduction capability of the
induced phage in this manuscript. Therefore, we have removed all mention of transducing
phage throughout the manuscript, except for their theorized role in the Discussion. As the
capability of these RpoS-induced phage to participate in transduction is our working model,
we have retained this hypothesis in the Discussion.

Additional Comments:

1. Throughout the manuscript. The idea that the phage observed are transducing phage is
intriguing but not supported here by data to corroborate it. The regulation of the phage
induction by BBD18 levels decreasing, with RpoS levels increasing, is itself most impressive.
The possibility that the transducing phage are encoded independent of the cp32 prophage
is feasible. Changing the terminology to phage induction without mention of its transducing
potential would seem appropriate.

Thank you for this suggestion; we have begun to investigate the role of RpoS-dependent
phage induction during horizontal gene transfer in *Borrelia*, but this is beyond the scope of
the current manuscript. Therefore, we removed "transducing" on line 30 of the **Abstract**,
changed the term "transducing" to "endogenous" on line 68 of the **Introduction**, and
removed "transducing" to state that phage "induction" was evidenced on line 535 of the
**Conclusion**.

2. Introduction, lines 81-84. This final statement hints at a demonstration within the
manuscript that phage-based lateral gene transfer is being evaluated. It is not and, thus, this
statement is best limited to the Discussion where such issues are generally presented.

We have revised the last sentence of the **Introduction** to indicate that it is speculative in
nature (lines 79-83) and present our theories on the role of transducing phage in the
**Discussion** (lines 502-519, 534-540).

3. Results, lines 214-216 and Figure 5 and Fig. 7. The observation that the copy number of
lp28-2 is also increased with the cp32s suggesting that lp28-2 encodes a separate
endogenous prophage. In Fig. 5, a similar phenomenon is seen for lp28-1 albeit without
statistical significance. Whether this represents an additional prophage or not, or is an
aberration, its increased copy number should be mentioned.

This is an accurate statement; therefore, we added the clause "although not significant, an
increase in copy number of lp28-1 was also evident" to Lines 276-277.

REVIEWERS' COMMENTS

Reviewer #2 (Remarks to the Author):

This is a beautifully performed and presented study.

Reviewer #3 (Remarks to the Author):

The initial manuscript was a solid report and is improved with the modifications made. The study is very interesting and is an important contribution to vector-borne microbial pathogenesis. I have only a couple of comments and one minor point for the authors to consider.

General Comments:

1. Introduction, lines 79-83. This final statement remains speculative. The way the sentence is worded it sounds as if the manuscript will demonstrate horizontal gene transfer. Qualifying it as what the data herein shows, or in the context of their current model, would be preferred.
2. Figure 1. The schematic here shows SR0736, which is the *ittA* sRNA. The co-authors refer to the sRNA as *ittA* in the text numerous times. As such, SR0736 should also be referred to as *ittA* in the figure.

Minor point.

1. Throughout the manuscript; lines 212, 238, 245, 257, 260, 267 (twice), and 341. The term *dropS* should be corrected to use the Greek character delta in front of *rpoS*.

We would like to thank Reviewer #3 for their positive and constructive feedback. Please find the response to Reviewer #3 comments below:

The initial manuscript was a solid report and is improved with the modifications made. The study is very interesting and is an important contribution to vector-borne microbial pathogenesis. I have only a couple of comments and one minor point for the authors to consider.

General Comments:

1. Introduction, lines 79-83. This final statement remains speculative. The way the sentence is worded it sounds as if the manuscript will demonstrate horizontal gene transfer. Qualifying it as what the data herein shows, or in the context of their current model, would be preferred.

We have removed lines 81-83 to eliminate this speculative inference of the final statement.

2. Figure 1. The schematic here shows SR0736, which is the *ittA* sRNA. The co-authors refer to the sRNA as *ittA* in the text numerous times. As such, SR0736 should also be referred to as *ittA* in the figure.

We have changed Figure 1 so that the sRNA SR0736 is now designated as *ittA* as it is referred to throughout the manuscript.

Minor point.

1. Throughout the manuscript; lines 212, 238, 245, 257, 260, 267 (twice), and 341. The term dropS should be corrected to use the Greek character delta in front of *rpoS*.

These have been changed from drpoS to ΔrpoS .